# Operation-Level Early Stopping for Robustifying Differentiable NAS

**Shen Jiang**[*]    **Zipeng Ji**[*]    **Guanghui Zhu**[†]    **Chunfeng Yuan**

**Yihua Huang**

State Key Laboratory for Novel Software Technology, Nanjing University
{jiangshen, jizipeng}@smail.nju.edu.cn, {zgh, cfyuan, yhuang}@nju.edu.cn

## Abstract

Differentiable NAS (DARTS) is a simple and efficient neural architecture search method that has been extensively adopted in various machine learning tasks. Nevertheless, DARTS still encounters several robustness issues, mainly the domination of skip connections. The resulting architectures are full of parametric-free operations, leading to performance collapse. Existing methods suggest that the skip connection has additional advantages in optimization compared to other parametric operations and propose to alleviate the domination of skip connections by eliminating these additional advantages. In this paper, we analyze this issue from a simple and straightforward perspective and propose that the domination of skip connections results from parametric operations overfitting the training data while architecture parameters are trained on the validation data, leading to undesired behaviors. Based on this observation, we propose the operation-level early stopping (OLES) method to overcome this issue and robustify DARTS without introducing any computation overhead. Extensive experimental results can verify our hypothesis and the effectiveness of OLES.

## 1 Introduction

Neural architecture search (NAS) aims to automatically design high-performance neural architectures Zoph and Le [2016] for given tasks in a data-driven manner. Deep learning models designed by NAS have achieved state-of-the-art performance in a variety of tasks, such as image classification Termritthikun et al. [2021], Real et al. [2018], Pham et al. [2018], semantic segmentation Liu et al. [2019], graph learning Zhu et al. [2022a, 2023], and recommender systems Chen et al. [2022], Zhu et al. [2022b]. Among various NAS methods, differentiable architecture search (DARTS) Liu et al. [2018] significantly improves the efficiency of NAS through parameter-sharing and continuous relaxation. DARTS converts discrete architecture selections into continuous architecture parameters $\alpha$ and optimizes them via gradient-based optimizers.

Despite its simplicity and efficiency, DARTS still has several issues that make it difficult to outperform other NAS methods, even a simple random search Li and Talwalkar [2020], Yu et al. [2019], in many cases. The first issue is the *performance gap*. Due to the discrepancies between discrete architectures and continuous architecture parameters, the derived model shows non-negligible performance degradation Liu et al. [2018]. Additionally, DARTS suffers from serious *robustness problems* Nayman et al. [2019], Chen et al. [2019], Liang et al. [2019], Chu et al. [2020], Zela et al. [2019], Wang et al.

---

[*]Equal contribution.
[†]Corresponding author.

[2021]. In particular, during the search phase of DARTS, as the number of training epochs increases, the searched architectures will fill with parametric-free operations Chu et al. [2019a], such as skip connections or even random noise, leading to performance collapse. These issues waste a lot of computational resources and seriously hinder the application of DARTS.

The domination of skip connections is the major robustness issue of DARTS Wang et al. [2021]. Many studies have investigated why the domination of skip connections happens and how to alleviate this issue. Most of them believe that skip connections exhibit unfair advantages in gradient-based optimization during the search phase Liang et al. [2019], Chu and Zhang [2020], Chu et al. [2019a] or stabilize the supernet training by constructing residual blocks Chu et al. [2020]. To address this problem, they propose eliminating these unfair advantages Chu et al. [2019a] or separating the stabilizing role Chu et al. [2020], mainly drawing inspiration from ResNet He et al. [2015]. However, the generic optimization in ResNet differs fundamentally from the bi-level optimization in DARTS. Therefore, observations and theories based on uni-level optimization are insufficient to support the claim that skip connections still have unfair advantages in DARTS. The work Wang et al. [2021] analyzes this problem from the perspective of architecture selection and proposes that the magnitude of the architecture parameter may not reflect the strength of the operation. Based on this assumption, a perturbation-based architecture selection method is proposed. However, this idea conflicts with the fundamental motivation of DARTS, which utilizes continuous relaxation to solve the combinatorial optimization problem approximately. Xie et al. [2021] suggests that overfitting in bi-level optimization is the primary reason for the catastrophic failure of DARTS and alayzes it from a unified theoretical framework, which is similar to our conjecture.

In contrast to the above sophisticated assumptions, in this paper, we take a simpler and more straightforward perspective to consider the reasons for the domination of skip connections. We observe that the robustness problem of DARTS is caused by overfitting. Specifically, the parameters of operations (e.g., convolution) in the supernet overfit the training data and gradually deviate from the validation data. Meanwhile, the architecture parameters are trained on the validation data, causing the advantages of parametric operations to weaken progressively and ultimately leading to the domination of parametric-free operations. It is evident that the supernet is always overparameterized. Additionally, the architecture parameters also affect the supernet training, with operations that exhibit advantages in the early stage of search being more prone to overfitting.

Based on the above observations, we further propose an operation-level early stopping (OLES) [1] method to address the domination of skip connections. The OLES method monitors each operation in the supernet to determine if it tends to overfit the training data and stops training the operation when overfitting occurs. Specifically, we utilize the gradient matching (GM) method Hu et al. [2022] to compare the gradient directions of operation parameters on the training data and the validation data. If the difference between them remains large in multiple consecutive iterations, the operation will stop training. This approach not only alleviates the domination of skip connections and but also sharpens the distribution of architecture parameters with more search iterations, thereby reducing the performance gap caused by discretization. Moreover, our method only needs to maintain the directions of operation gradients. Due to only minimal modifications to the original DARTS, the additional overhead is negligible.

In summary, we make the following contributions:

- We analyze the robustness issue of DARTS from a new perspective, focusing on the overfitting of operations in the supernet.

- We propose the operation-level early stopping method to address the domination of skip connections in DARTS with negligible computation overheads.

- Extensive experiments demonstrate the effectiveness and efficiency of our method, which can achieve state-of-the-art (SOTA) performance on multiple commonly used image classification datasets.

---

[1]Open-source code can be found at https://github.com/PasaLab/oles.

## 2 Background and Related Work

### 2.1 Neural Architecture Search

Over the years, a variety of NAS methods have been proposed, including evolutionary-based Termrit-thikun et al. [2021], Real et al. [2018], Guo et al. [2020], reinforcement-learning-based Zoph and Le [2016], Pham et al. [2018], and gradient-based Liu et al. [2018], Xie et al. [2018], Cai et al. [2018] methods. They have achieved great success in many computer vision (CV) and natural language processing (NLP) So et al. [2019] tasks. As a pioneering work, NAS-RL Zoph and Le [2016] uses an RNN controller to generate architectures and employs the policy gradient algorithm to train the controller. However, this method suffers from low search efficiency. To improve the search efficiency, the cell-based micro search space Zoph et al. [2017] and weight-sharing mechanism Pham et al. [2018] have been proposed. DARTS Liu et al. [2018], which also takes advantage of weight-sharing, utilizes the gradient-based method to optimize architecture parameters and model weights alternatively. Recently, train-free NAS methods Chen et al. [2021a], Li et al. [2023] that utilize proxy metrics to predict the ranking or test performance of architectures without training have been proposed to further promote search efficiency.

### 2.2 Preliminaries of Differentiable Neural Architecture Search

We will now provide a detailed review of DARTS. The original DARTS Liu et al. [2018] relies on a cell-based micro search space, where each cell contains $N$ nodes and $E$ edges organized in a directed acyclic graph (DAG). Each node represents a feature map $x^{(i)}$, and each edge is associated with an operation $o \in \mathcal{O}$, where $\mathcal{O}$ denotes the set of candidate operations (e.g., skip connect, sep conv 3x3, etc.). During architecture search, instead of applying a single operation to a specific node, continuous relaxation is applied to relax the categorical choice of a specific operation to a mixture of candidate operations, i.e., $\bar{o}^{(i,j)}(x) = \sum_{o \in \mathcal{O}} \frac{\exp(\alpha_o^{(i,j)})}{\sum_{o' \in \mathcal{O}} \exp(\alpha_{o'}^{(i,j)})} o(x)$. $\alpha = \{\alpha^{(i,j)}\}$ serves as the architecture parameters. DARTS then jointly optimizes architecture parameters $\alpha$ and model weights $w$ with the following bi-level objective via alternative gradient descent:

$$\min_{\alpha} \mathcal{L}_{val}(w^*, \alpha)$$
$$\text{s.t.} \quad w^* = \arg\min_{w} \mathcal{L}_{train}(w, \alpha).$$

Since DARTS follows the weight-sharing mechanism, we refer to the continuous relaxed network as the "supernet". At the end of the search phase, a discretization phase is performed by selecting operations with the largest $\alpha$ values to form the final architecture.

### 2.3 Robustifying DARTS

Despite its simplicity and efficiency, the undesired domination of skip connections is often observed in DARTS. Previous works have attempted to eliminate the domination of skip connections by diminishing their unfair advantages. DARTS+Liang et al. [2019] uses a fixed value to constrain the number of skip connections. Progressive DARTSChen et al. [2019] adds a Dropout after each skip connection operation. DARTS−Chu et al. [2020] introduces an auxiliary skip connection to separate the unfair advantage of the skip connection by analyzing it from the perspective of ResNet He et al. [2015]. Another strategy is to remove the exclusive competition of operation selection in DARTS. FairDARTSChu et al. [2019a] replaces the *Softmax* relaxation of DARTS with *Sigmoid* and offers an independent architecture parameter for each operation. NoisyDARTSChu and Zhang [2020] injects unbiased noise into the candidate operations to ensure that the good ones win robustly. However, DARTS+PT Wang et al. [2021] believes that the magnitude of the architecture parameter is not a good indicator of operation strength and proposes a perturbation-based architecture selection method.

Our work is also related to GM-NAS Hu et al. [2022], which falls in the field of Few-Shot NAS Zhao et al. [2021]. Although our method and GM-NAS both employ the GM score as a mathematical tool, they are entirely distinct lines of work. GM-NAS argues that due to coupled optimization between child architectures caused by weight-sharing, One-Shot supernet's performance estimation could be inaccurate, leading to degraded search results. As a result, GM-NAS proposes to reduce the level of weight-sharing by splitting the One-Shot supernet into multiple separated sub-supernets.

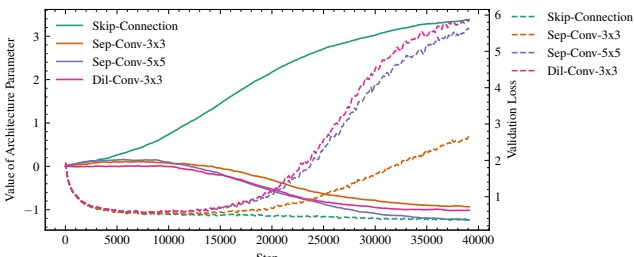 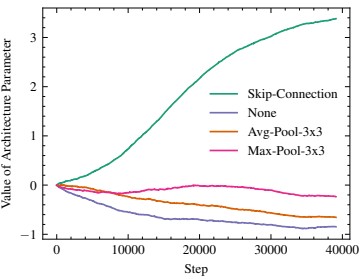

(a) The architecture parameters and validation losses of different operations. The solid line represents the value of architecture parameter, while the dotted line represents the validation loss.

(b) The architecture parameters of different parametric-free operations.

Figure 1: The motivational experiment to vefiry the domination of skip connections is due to the overfitting of operations.

GM-NAS utilizes GM scores to make splitting decisions, determining whether a module should be shared among child architectures. The GM scores in GM-NAS are computed based on the gradient information of different child architectures on shared parameters. In contrast, we aim to address the problem of skip connection donimation from a totally new perspective and employ the GM score as an indicator for early stopping, preventing operation parameter overfitting. The GM scores, in our approach, are calculated using the gradient information of parameters on training and validation data.

## 3 Motivation and Methodology

In this section, we argue that the domination of skip connections is due to the overfitting of operations in the supernet and demonstrate this hypothesis through a simple motivational experiment. Based on our observations, we propose an operation-level early stopping method to address this issue.

### 3.1 The Domination of Skip-Connection is Due to Overfitting

In DARTS, we observe that skip connections dominate due to the overfitting of operations in the supernet to the training data, causing them to increasingly deviate from the distribution of the validation data. However, the architecture parameters are trained on the validation data, thus operations are selected based on their validation performance. As the number of training iterations increases, the parametric operations overfit more seriously, and the parametric-free operations gradually dominate.

To verify the above conjecture, we present empirical evidence on the relationship between the overfitting of operations in the supernet and architecture parameters. We retain only architectural parameters on one edge in DARTS' search space and set all other architectural parameters to $0$ to ensure that they will not be trained. In Figure 1, we record the architecture parameter and the validation loss with the corresponding output of each operation in this edge. From Figure 1a, we can observe that the validation losses of all parametric operations grow even faster as the training steps increase, and the architecture parameter gradually decreases, exhibiting an obvious negative correlation with the validation loss. Moreover, Figure 1bshows that the skip connection easily obtains higher architecture parameters, exhibiting advantages over other parametric-free operations. This is mainly due to the fact that the skip connection can always retain more information from the input.

Most existing works suggest that the domination of skip connections is due to their unfair advantage. To investigate this claim, we follow the settings of FairDARTS Chu et al. [2019a] and replace the *Softmax* relaxation in DARTS with the *Sigmoid* relaxation. The unfair advantage of skip connections will be eliminated by removing the exclusive competition among operations. We then rerun the experiment and present the results in Figure 2, which shows that even after removing the exclusive competition, the architecture parameters still exhibit a negative correlation with the validation losses, similar to that in Figure 1a. It illustrates that the unfair advantage of skip connections is not the fundamental reason for their domination. Instead, it is the effect of the overfitting of operations in the supernet on the training of architecture parameters.

## 3.2 Operation-Level Early Stopping

Based on the above observations, we propose an operation-level early stopping method to alleviate the domination of skip connections and enhance the robustness of DARTS. During the search phase, the architecture parameters have an effect on the supernet training, resulting in different operations with different architecture parameters being trained differently. Operations that exhibit advantages in the early stage will get more attention and are more prone to overfitting. Moreover, the operation is the basic unit of

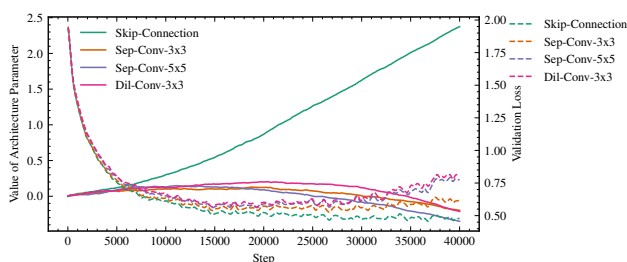

Figure 2: The architecture parameters and validation losses of different operations in FairDARTS.

choice in DARTS, which selects operations to form the final architecture. Therefore, we perform early stopping for each operation. If early stopping can be performed for each operation during the supernet training, when the operation tends to overfit, the overfitting problem can have less impact on the training of architecture parameters, and the supernet can also be fully trained.

Overfitting occurs when a model is too complex and fits the training data distribution too closely, resulting in poor performance on the validating data. When a model is overfitting, the training loss continues to decrease while the validation loss increases, indicating that the optimization direction (i.e., the direction of gradients) of the model parameters on the training data will be inconsistent with that on the validation data. To identify overfitting, we propose using the gradient direction of the operation parameters. If the gradients on the training and validation data differ significantly in direction, we consider the operation to tend to overfit the training data.

---

**Algorithm 1** DARTS with operation-level early stopping

---

**Require:** Supernet weights $w$; architecture parameters $\alpha$; number of search iterations $I$; number of consecutive iterations for GM $M$; threshold of GM score for verifying overfitting $\xi$.
**Ensure:** Searched architectures.
 1: Construct a supernet $A$;
 2: **for** each $i \in [1, I]$ **do**
 3:     Get a batch of training data $B_{train}$;
 4:     Update weights $w$ by $\nabla_w \mathcal{L}_{train}(w, \alpha, B_{train})$;
 5:     Record the training gradient of each operation $g_{train}^o$;
 6:     Get a batch of validation data $B_{val}$;
 7:     Update parameters $\alpha$ by $\nabla_\alpha \mathcal{L}_{val}(w, \alpha, B_{val})$;
 8:     Record the validation gradient of each operation $g_{val}^o$;
 9:     Compute the gradient matching score over consecutive $M$ iterations based on Eq. 1.
10:     Stop the gradient of an operation $o$ if $GM(g_{train}^o, g_{val}^o) < \xi$
11: **end for**
12: Derive the final architecture based on learned $\alpha$.

---

Concretely, for an operation $o$, let $P_o$ denote the parameters of $o$, and $g_{train}^o$ and $g_{val}^o$ denote the gradients of $P_o$ on a training batch and a validation batch, respectively. We calculate the cosine similarity between $g_{train}^o$ and $g_{val}^o$ to quantify the differences in the direction of the operations' gradients as follows:

$$GM(g_{train}^o, g_{val}^o) = \frac{\sum_{m=1}^{M} \text{COS}(g_{train}^o, g_{val}^o)}{M},$$

$$\text{where} \quad \text{COS}(g_{train}^o, g_{val}^o) = \frac{g_{train}^o \cdot g_{val}^o}{|g_{train}^o||g_{val}^o|}, \tag{1}$$

$$g_{train}^o = \frac{\partial \mathcal{L}_{train}(w, \alpha, B_{train})}{\partial P_o}, g_{val}^o = \frac{\partial \mathcal{L}_{val}(w, \alpha, B_{val})}{\partial P_o},$$

where $B_{train}$ and $B_{val}$ denote batches of training data and validation data, respectively. The GM score is computed as the average cosine similarity between training gradients and validation gradients

over $M$ consecutive steps. The gradient matching score here is similar to that in GM-NAS Hu et al. [2022]. When the GM score falls below a predefined threshold of $\xi \in (0, 1)$ within $M$ steps, the operation is considered to be overfitting. Consequently, if $GM < \xi$ for an operation $o$, the parameters of this operation will not be updated during subsequent supernet training. And the remaining operations and architecture parameters will be trained as usual.

Through the operation-level early stopping method, the overfitting of operations in the supernet can be alleviated. Benefiting from the attribute that the DARTS search process involves alternating between training the supernet on the training data and training architecture parameters on the validation data, we only need to maintain the gradient of each operation on a training data batch and a validation data batch in one iteration, with negligible additional computational overhead. As a result, OLES can alleviate the domination of skip connections, making DARTS more robust. This allows for training DARTS with more iterations to obtain better architectures and sharper architecture parameters, resulting in a smaller discretization gap and improving performance. The algorithm for DARTS with operation-level early stopping can be found in Algorithm 1.

## 4  Experiments

### 4.1  Search Spaces and Experimental Settings

To verify the effectiveness of OLES, we conduct experiments on three different search spaces: the standard DARTS' search space, NAS-Bench-201, and MobileNet-like search space. We keep all experimental settings the same as the original in each search space and only use the *first-order* optimization, as our method only needs to freeze the operation parameters during the supernet training. In addition to the standard hyperparameters in DARTS, our method introduces an additional hyperparameter, i.e., the overfitting threshold $\xi$, to stop operation training. We determine the threshold $\xi$ by averaging the cosine similarity over 20 iterations for 30 randomly initiated architectures in each search space. And the gradient matching (GM) score is dynamically computed by averaging over every 20 iterations throughout the entire training process to reduce variance. Following Algorithm 1, we stop updating parameters of the corresponding operation $o$ when $GM(g^o_{train}, g^o_{val})$ is lower than the threshold $\xi$.

### 4.2  Performance on DARTS' CNN Search Space

#### 4.2.1  CIFAR-10

We conduct NAS on CIFAR-10 using DARTS' search space and then transfer the searched architectures to CIFAR-100. The threshold $\xi$ is set to $0.3$. As shown in Table 1, our method OLES consistently outperforms other recent SOTA NAS methods. OLES achieves a top-1 test error of $2.30\%$ on CIFAR-10 with nearly the same search cost as DARTS of $0.4$ GPU-days. Moreover, the robustness of OLES is also guaranteed, as evidenced by the excellent average result of three independent runs. When the architectures searched on CIFAR-10 are transferred to CIFAR-100, our method still achieves a competitive test error of $16.30\%$. This indicates that the architectures searched by OLES are transferable and expressive, and the architecture parameters do not overfit the CIFAR-10 dataset.

#### 4.2.2  CIFAR-100

We conduct NAS on CIFAR-100, and the results are presented in Table 2. The threshold $\xi$ is set to $0.4$. OLES still exhibits significant advantages, improving the test error of DARTS from $20.58\%$ to $17.30\%$ without incurring extra search costs. It demonstrates that our method can adapt to datasets of different sizes and consistently improve the performance of DARTS. OLES not only consistently outperforms other methods that focus on the skip-connection domination issue, but also surpasses more sophisticated NAS methods (e.g., GM+DARTS Hu et al. [2022]) on CIFAR-10 and CIFAR-100 without requiring extensive code modifications or introducing additional computational overheads.

#### 4.2.3  Transfer to ImageNet

To verify the transferability of architectures discovered by OLES, we transfer the best architecture derived from CIFAR-10 to ImageNet. Following the setting of DARTS, we train the searched

Table 1: Comparison with state-of-the-art NAS methods on CIFAR-10, and the transferred results on CIFAR-100 are also reported. $\diamond$: reported by Chen et al. [2019]. $*$: produced by rerunning the published code. $\ddagger$: transfer architectures discovered on CIFAR-10 to CIFAR-100.

| NAS method | Test Err(%) | | Params (M) | Search Cost (GPU-days) | Search Method |
|---|---|---|---|---|---|
| | CIFAR-10 | CIFAR-100 | | | |
| DenseNet-BC Huang et al. [2016] | 3.46 | - | 25.6 | - | Manual |
| ResNet + CutOutHe et al. [2015] | 4.61 | 17.8 | 1.7 | - | Manual |
| NASNet-A + CutOut Zoph et al. [2017] | 2.65 | - | 3.3 | 1800 | RL |
| AmoebaNet-A Real et al. [2018] | $3.34 \pm 0.06$ | - | 3.2 | 3150 | Evolution |
| AmoebaNet-B Real et al. [2018] | $2.55 \pm 0.05$ | - | 2.8 | 3150 | Evolution |
| PNAS Liu et al. [2017] | $3.41 \pm 0.09$ | - | 3.2 | 225 | MDL |
| ENAS + CutOut Pham et al. [2018] | 2.89 | - | 4.6 | 0.5 | RL |
| DARTS(1st) + CutOutLiu et al. [2018] | $3.00 \pm 0.14$ | $17.76^{\diamond}$ | 3.3 | 0.4 | Gradient |
| DARTS(2nd) + CutOutLiu et al. [2018] | $2.85 \pm 0.08*$ | $17.54^{\diamond}$ | 3.4 | 0.4 | Gradient |
| P-DARTS Chen et al. [2019] | 2.50 | 16.55 | 3.4 | 0.3 | Gradient |
| R-DARTS(L2) Zela et al. [2019] | $2.95 \pm 0.21$ | - | - | 1.6 | Gradient |
| FairDARTS Chu et al. [2019a] | 2.54 | - | 2.8 | 0.4 | Gradient |
| DARTS- Chu et al. [2020] | $2.59 \pm 0.08$ | - | $3.5 \pm 0.13$ | 0.4 | Gradient |
| SGAS(1st) Li et al. [2020] | 2.39 | - | 3.8 | 0.25 | Gradient |
| DARTS+PT Wang et al. [2021] | $2.48(2.61 \pm 0.08)$ | $19.05^{\ddagger}$ | 3.0 | 0.8 | Gradient |
| DrNAS Chen et al. [2021b] | $2.46 \pm 0.03$ | - | 4.1 | 0.6 | Gradient |
| $\beta$-DARTS Ye et al. [2022] | $2.53 \pm 0.08$ | $16.24 \pm 0.22$ | $3.75 \pm 0.15$ | 0.4 | Gradient |
| GM+DARTS(1st) Hu et al. [2022] | 2.35 | 16.45 | 3.7 | 1.1 | Gradient |
| **OLES** | **2.30(2.41 $\pm$ 0.11)** | 16.30(16.35 $\pm$ 0.05) | 3.4 | 0.4 | Gradient |

Table 2: Comparison with NAS methods on CIFAR-100. $\diamond$: reported by Dong and Yang [2019]. $\dagger$: reported by Zela et al. [2019]. $*$: produced by rerunning the published code.

| NAS method | Test Err (%) | Params (M) | Search Cost (GPU-days) | Search Method |
|---|---|---|---|---|
| ResNet + CutOutHe et al. [2015] | $22.10^{\diamond}$ | 1.7 | - | Manual |
| PNAS Liu et al. [2017] | $19.53^{\diamond}$ | 3.2 | 150 | MDL |
| ENAS + CutOut Pham et al. [2018] | $19.43^{\diamond}$ | 4.6 | 0.45 | RL |
| DARTS Liu et al. [2018] | $20.58 \pm 0.44^{\dagger}$ | 3.4 | 0.4 | Gradient |
| GDAS Dong and Yang [2019] | 18.38 | 3.4 | 0.2 | Gradient |
| P-DARTS Chen et al. [2019] | $17.46*$ | 3.6 | 0.3 | Gradient |
| R-DARTS(L2)Zela et al. [2019] | $18.24*$ | - | 1.6 | Gradient |
| DARTS- Chu et al. [2020] | $17.51 \pm 0.25$ | 3.3 | 0.4 | Gradient |
| PR-DARTS Zhou et al. [2020] | $20.10*$ | 3.6 | 0.17 | Gradient |
| DARTS+PT Wang et al. [2021] | $18.78*$ | 3.4 | 0.8 | Gradient |
| $\beta$-DARTSYe et al. [2022] | $17.33*$ | $3.83 \pm 0.08$ | 0.4 | Gradient |
| GM+DARTSHu et al. [2022] | $17.42*$ | 3.6 | 1.3 | Gradient |
| **OLES** | **17.30** | 3.4 | 0.4 | Gradient |

architecture from scratch for 250 epochs, and the results are presented in Table 3. Our method achieves $24.5\%$ top-1 test error. Compared to other NAS methods, the architectures derived from OLES still exhibit competitive performance. Notably, we do not introduce any additional training techniques such as SE modules and Swish, but simply follow the original DARTS setting to transfer the architectures. Therefore, the performance of OLES on ImageNet can be further enhanced by incorporating these training tricks.

### 4.3 Performance on NAS-Bench-201

NAS-Bench-201 Dong and Yang [2020] provides a unified benchmark for analyzing various up-to-date NAS algorithms. It contains 4 internal nodes with 5 operations (i.e., Zero, Skip Connection, $1 \times 1$ Conv, $3 \times 3$ Conv, $3 \times 3$ AvgPol). NAS-Bench-201 offers a similar cell-based search space that comprised a total of 15625 unique architectures. The architectures are trained on three datasets (i.e., CIFAR-10, CIFAR-100, and ImageNet16-120). The comparison results are shown in Table 4, and the threshold $\xi$ is set to 0.7. DARTS performs extremely poorly on NAS-Bench-201. We suppose that it is due to the fact that the search space of NAS-Bench-201 is extremely compact, contains few parametric operations, and the number of operation parameters varies widely, which makes it difficult for DARTS to adapt to NAS-Bench-201. OLES significantly outperforms DARTS on all datasets and is competitive among other NAS methods. It demonstrates the generality of OLES, which can adapt to different search spaces and datasets.

Table 3: Comparison with NAS methods on ImageNet. ⋄: reported by Chu et al. [2019a]. Other results are from their original papers. DARTS- and NoisyDARTS-A are directly searched on ImageNet and use SE modules and Swish to improve performance. Other methods in the bottom block are searched on CIFAR-10 and then transferred to ImageNet.

| NAS method | Test Err (%) | | Params (M) | Search Cost (GPU-days) | Search Method |
|---|---|---|---|---|---|
| | Top-1 | Top-5 | | | |
| Inception-v1 Szegedy et al. [2015] | 30.2 | 10.1 | 6.6 | - | Manual |
| MobileNet-V2 Sandler et al. [2018] | 28.0 | - | 3.4 | - | Manual |
| ShuffleNet Zhang et al. [2017] | 26.3 | - | ∼5 | - | Manual |
| NASNet-A Zoph et al. [2017] | 26.0 | 8.4 | 5.3 | 1800 | RL |
| AmoebaNet-A Real et al. [2018] | 25.5 | 8.0 | 5.1 | 3150 | Evolution |
| AmoebaNet-B Real et al. [2018] | 26.0 | 8.5 | 5.3 | 3150 | Evolution |
| PNAS Liu et al. [2017] | 25.8 | 8.1 | 5.1 | 225 | SMBO |
| MnasNet-92 Tan et al. [2018] | 25.2$^\diamond$ | 8.0$^\diamond$ | 4.4 | 1667 | RL |
| DARTS(2nd) + CutOutLiu et al. [2018] | 26.9 | 8.7 | 4.7 | 0.4 | Gradient |
| SNAS Xie et al. [2018] | 27.3 | 9.2 | 4.3 | 1.5 | Gradient |
| GDAS Dong and Yang [2019] | 25.0 | 8.5 | 5.3 | 0.2 | Gradient |
| P-DARTS Chen et al. [2019] | 24.4 | 7.4 | 5.1 | 0.3 | Gradient |
| PC-DARTS Xu et al. [2019] | 25.1 | 7.8 | 5.3 | 3.8 | Gradient |
| FairDARTS-B Chu et al. [2019a] | 24.9 | 7.5 | 4.8 | 0.4 | Gradient |
| NoisyDARTS-A Chu and Zhang [2020] | 22.1 | 6.0 | 5.5 | 12 | Gradient |
| DARTS- Chu et al. [2020] | 23.8 | 6.1 | 5.5 | 4.5 | Gradient |
| DARTS+PT Wang et al. [2021] | 25.5 | 8.0 | 4.6 | 0.8 | Gradient |
| DrNAS Chen et al. [2021b] | 23.7 | 7.1 | 5.7 | 4.6 | Gradient |
| $\beta$-DARTSYe et al. [2022] | 23.9 | 7.0 | 5.5 | 0.4 | Gradient |
| GM + DARTS(2nd) Hu et al. [2022] | 24.5 | 7.3 | 5.1 | 2.7 | Gradient |
| **OLES** | 24.5 | 7.4 | 4.7 | 0.4 | Gradient |

Table 4: Results on NAS-Bench201.

| NAS method | CIFAR-10 | | CIFAR-100 | | ImageNet16-120 | |
|---|---|---|---|---|---|---|
| | valid | test | valid | test | valid | test |
| DARTS(2nd) Liu et al. [2018] | $39.77 \pm 0.00$ | $54.30 \pm 0.00$ | $15.03 \pm 0.00$ | $15.61 \pm 0.00$ | $16.43 \pm 0.00$ | $16.32 \pm 0.00$ |
| SNAS Xie et al. [2018] | $90.10 \pm 1.04$ | $92.77 \pm 0.83$ | $69.69 \pm 2.39$ | $69.34 \pm 1.98$ | $42.84 \pm 1.79$ | $43.16 \pm 2.64$ |
| DSNAS Hu et al. [2020] | $89.96 \pm 0.29$ | $93.08 \pm 0.13$ | $30.87 \pm 16.40$ | $31.01 \pm 16.38$ | $40.61 \pm 0.09$ | $41.07 \pm 0.09$ |
| PC-DARTS Xu et al. [2019] | $89.96 \pm 0.15$ | $93.41 \pm 0.30$ | $67.12 \pm 0.39$ | $67.48 \pm 0.89$ | $40.83 \pm 0.08$ | $41.31 \pm 0.22$ |
| DARTS- Chu et al. [2020] | $91.03 \pm 0.44$ | $93.48 \pm 0.40$ | $71.36 \pm 1.51$ | $71.53 \pm 1.51$ | $44.87 \pm 1.46$ | $45.12 \pm 0.82$ |
| iDARTS Zhang et al. [2021a] | $89.86 \pm 0.60$ | $93.58 \pm 0.32$ | $70.57 \pm 0.24$ | $70.83 \pm 0.48$ | $40.38 \pm 0.59$ | $40.89 \pm 0.68$ |
| RLNAS(random label)Zhang et al. [2021b] | $89.94 \pm 0.00$ | $93.35 \pm 0.00$ | $70.98 \pm 0.00$ | $70.71 \pm 0.00$ | $43.86 \pm 0.00$ | $43.70 \pm 0.00$ |
| DrNASChen et al. [2021b] | $91.55 \pm 0.00$ | $94.36 \pm 0.00$ | $73.49 \pm 0.00$ | $73.51 \pm 0.00$ | $46.37 \pm 0.00$ | $46.34 \pm 0.00$ |
| $\beta$-DARTS Ye et al. [2022] | $91.55 \pm 0.00$ | $94.36 \pm 0.00$ | $73.49 \pm 0.00$ | $73.51 \pm 0.00$ | $46.37 \pm 0.00$ | $46.34 \pm 0.00$ |
| GM + DARTSHu et al. [2022] | $91.03 \pm 0.24$ | $93.72 \pm 0.12$ | $71.61 \pm 0.62$ | $71.83 \pm 0.97$ | $42.19 \pm 0.00$ | $42.60 \pm 0.00$ |
| **OLES** | $90.88 \pm 0.10$ | $93.70 \pm 0.15$ | $70.56 \pm 0.28$ | $70.40 \pm 0.22$ | $44.17 \pm 0.49$ | $43.97 \pm 0.38$ |

## 4.4 Performance on MobileNet-like Search Space

As described in FairDARTS Chu et al. [2019a], the domination of skip connections also exists in the MobileNet-like search space. The supernet is built on MobileNetV2 Sandler et al. [2018] and comprises 21 choice blocks, each with 7 candidate operations. As the MobileNet search space is not naturally designed for DARTS, we make minor modifications to the MobileNet search space as FairDARTS and perform NAS on the ImageNet dataset. The threshold $\xi$ is set to $0.4$. The results are summarized in Table 5, and the corresponding searched architecture is visualized in Appendix D.1. We can observe that OLES can also achieve highly competitive performance in the MobileNet search space.

Table 5: Results on ImageNet using the MobileNet-like search space.

| NAS method | Top-1 | Top-5 | Params(M) |
|---|---|---|---|
| PloxylessNAS Cai et al. [2018] | 24.9 | 7.5 | 7.1 |
| FBNet-C Wu et al. [2018] | 25.1 | 7.9 | 4.4 |
| FairNAS-A Chu et al. [2019b] | 24.7 | 7.6 | 4.6 |
| FairDARTS-D Chu et al. [2019a] | 24.4 | 7.4 | 4.3 |
| RLNAS Zhang et al. [2021b] | 24.4 | 7.4 | 5.3 |
| GM+ProxylessNAS Hu et al. [2022] | 23.4 | 7.0 | 4.9 |
| **OLES** | 24.7 | 7.6 | 4.7 |

# 5 In-Depth Analysis

## 5.1 Different Gradient Direction Similarity Measures

Table 6: Test accuracy(%) with different gradient direction similarity measures on NAS-Bench-201.

| Dataset | Cosine Similarity | $L_2$ Distance | Direction Distance | Hard Threshold |
|---------|-------------------|----------------|--------------------|----------------|
| CIFAR-10 | **93.77** | 92.35 | 90.59 | 93.65 |
| CIFAR-100 | 70.11 | 69.39 | 67.40 | **70.19** |
| ImageNet | **44.23** | 42.38 | 38.28 | 43.86 |

As described in Section 3.2, we use cosine similarity to measure the difference in the operation's gradient directions between the training and validation data. To examine the effect of different gradient direction similarity measures for computing the gradient matching score, we compare cosine similarity with three other measures: $L_2$ distance, direction distance, and hard threshold. The mathematical definitions of the other three measures can be found in Appendix B. The direction distance directly computes the fraction of elements that have different directions in two gradients. And the hard threshold does not use the averaged value of $M$ iterations to compute the GM score but instead directly checks whether the direction distance can consistently be greater than a threshold in consecutive $M$ iterations. As shown in Table 6, cosine similarity performs better than the other three methods. Therefore, we adopt cosine similarity as the gradient direction similarity measure of OLES.

## 5.2 Gradient Matching Score

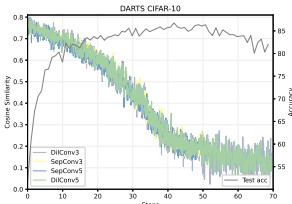
(a) Operations' cosine similarity and test accuracy on CIFAR-10.

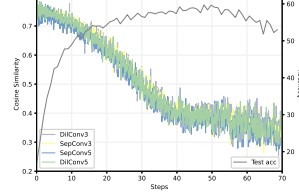
(b) Operations' cosine similarity and test accuracy on CIFAR-100.

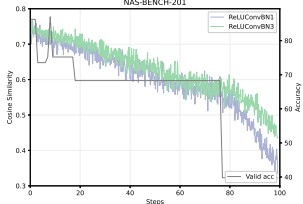
(c) Operations' cosine similarity and test accuracy on CIFAR-10 (NAS-Bench-201).

Figure 3: Trajectories of the cosine similarity of parametric operations and test accuracy during the training of DARTS.

In Figure 3, we present the trajectories of consine similarity between gradients on training data and validation data during the DARTS training. With the training process of DARTS, the cosine similarities of parametric operations gradually decrease, which indicates that the cosine similarity can reflect the overfitting of operations. And different parametric operations show similar trajectories. Consequently, we set an unified GM score threshold for all parametric operations. Moreover, when the number of training iterations continuously increases (>50), DARTS exhibits a noticeable decline in test performance. Figure 3c further corroborates this finding, as the problem of performance collapse due to the skip connection domination issue more easily appears in NAS-Bench-201. As a result, it manifests the test performance declines at an earlier stage.

## 5.3 Training With Longer Epochs

As mentioned in Bi et al. [2019] and Chu et al. [2020], training DARTS with longer epochs enables a better convergence of the supernet and architecture parameters. However, as shown in Figure 4a and 4b, training standard DARTS with longer epochs exhibit serious performance collapse with the domination of skip connections. The skip connection rapidly dominates after 50 epochs on CIFAR-10 and takes longer epochs on CIFAR-100. And we also find that other methods (e.g., FairDARTS, and DARTS-PT) fail with longer training epochs as well. In contrast, as shown in Figure 4c and 4d, OLES survives even after 200 epochs. The number of skip connections in OLES stays stable after a specific number of epochs since most parametric operations are frozen. It demonstrates that OLES is robust and can actually address the domination of skip connections. More experimental results and searched architectures with longer training epochs can be found in Appendix C.2 and Appendix D.2.

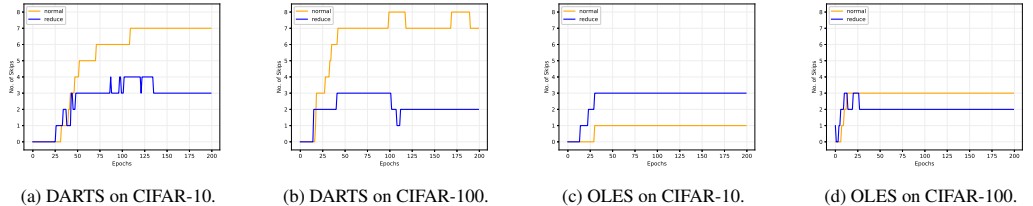

| (a) DARTS on CIFAR-10. | (b) DARTS on CIFAR-100. | (c) OLES on CIFAR-10. | (d) OLES on CIFAR-100. |
|---|---|---|---|

Figure 4: Trajectories of the numbers of skip connections in the normal cell and the reduction cell on CIFAR-10 and CIFAR-100.

## 5.4 Comparison of Kendall Rank Correlation Coefficients

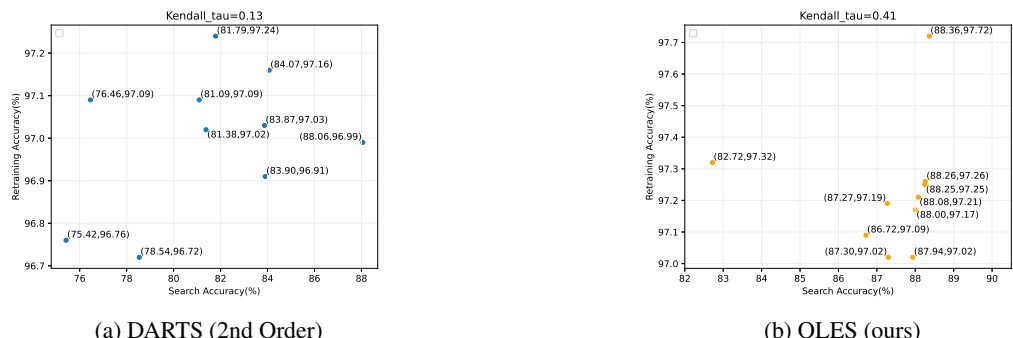

(a) DARTS (2nd Order)                                   (b) OLES (ours)

Figure 5: Comparison of search-retraining Kendall rank correlation coefficients $\tau$. The results are derived from 10 randomly selected architectures from a single run of DARTS and OLES.

Table 7: Kendall-$\tau$ coefficients of architecture ranks based on searching and retraining metrics. ∗: sample 4 excellent architectures in 4 rounds. Other methods randomly select 10 architectures from a single run.

| RandomNAS∗ | DARTS | GDAS∗ | SGAS | OLES |
|---|---|---|---|---|
| 0.0909 | 0.13 | -0.1818 | 0.42 | 0.41 |

Figure 5 indicates that the early stopping mechanism does not hurt the ranking performance of DARTS. Additionally, we compare the Kendall rank correlation coefficients with other NAS methods, including RandomNAS, DARTS, GDAS, and SGAS, as shown in Table 7.

Notably, the Kendall coefficient of OLES aligns closely with that of SGAS Li et al. [2020], which aims to alleviate the effect of the degenerate search-retraining correlation problem. The results demonstrate that by mitigating operation parameter overfitting, differentiable NAS is able to focus on the potential of architectures themselves, thus enhancing the correlation between search metrics and the architectures discovered.

## 6 Conclusion

This paper attempts to find the fundamental reason for the domination of skip connections in DARTS from the new perspective of overfitting of operations in the supernet. We verified our conjecture through motivational experiments and proposed the operation-level early stopping (OLES) method to solve the domination of skip connections. OLES monitors the gradient direction of each operation and determines whether it is overfitting based on the difference between the gradient directions on the training data and the validation data. OLES can solve the domination of skip connections with negligible computational overheads without imposing any limitations. Moreover, OLES achieves state-of-the-art results of $2.30\%$ test error on CIFAR-10 and $17.30\%$ test error on CIFAR-100, meanwhile reporting superior performance when transferred to ImageNet and on search spaces of NAS-Bench-201 and MobileNet as well.

## Acknowledgment

This work was supported by the National Natural Science Foundation of China (#62102177), the Natural Science Foundation of Jiangsu Province (#BK20210181), the Key R&D Program of Jiangsu Province (#BE2021729), and the Collaborative Innovation Center of Novel Software Technology and Industrialization, Jiangsu, China.

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

# A  The Selection of Parametric-Free Operations on Various Search Spaces

Zela et al. [2019] designed four variants of the DARTS' search space (S1-S4) and pointed out skip connections tend to dominate compared with two other parametric-free operations, i.e., *zero* and *noise*. To investigate whether the robustness issue in DARTS arises from the domination of skip connections rather than other parametric-free operations, we extend the experimental settings in Zela et al. [2019] to seven search spaces, including four types of parametric-free operations( *avg_pool_-3×3, skip_connect, noise, zero*). These search spaces only differ from DARTS' original space in the number and types of operations on each edge. We follow the same experimental setup in Liu et al. [2018] and execute DARTS on each search space. Figures 6-8 and 13-15 display the results on S1-S3. We observe that skip connections tend to dominate more severely than other parametric-free operations. By designing different combinations of parametric-free operations in S4-S7 search spaces, Figures 9-12 and 16-19 show that skip connections also dominate under fair competition with other parametric-free operations when the searched cells contain more than one type of parametric-free operations. The corresponding performance for each search space is shown in Table 8. As a result, the domination of skip connections is the primary cause of the shallow architecture of DARTS with poor generalization and robustness.

- S1 consists of two operations: *skip_connect* and *sep_conv_3×3*.

- S2 consists of two operations: *avg_pool_3×3* and *sep_conv_3×3*.

- S3 consists of two operations: *noise* and *sep_conv_3×3*. The noise operation replaces each input value with a random Gaussian noise for $N(0, 1)$. Differentiable NAS algorithms generally avoid this operation because it hurts the performance of discretized architectures.

- S4 consists of three operations: *skip_connect*, *noise*, and *sep_conv_3×3*.

- S5 consists of three operations: *skip_connect*, *avg_pool_3×3*, and *sep_conv_3×3*.

- S6 consists of three operations: *skip_connect*, *zero*, and *sep_conv_3×3*. The zero operation replaces each input value with zeros.

- S7 consists of five operations: *skip_connect*, *avg_pool_3×3*, *noise*, *zero*, and *sep_conv_3×3*.

Table 8: Performance of DARTS on search spaces S1-S7.

| Search Space | CIFAR-10 | CIFAR-100 |
|:---:|:---:|:---:|
| S1 | 3.28 | 26.05 |
| S2 | 2.55 | 25.40 |
| S3 | 3.69 | 24.70 |
| S4 | 3.05 | 21.35 |
| S5 | 3.03 | 23.77 |
| S6 | 2.84 | 27.90 |
| S7 | 2.70 | 22.17 |

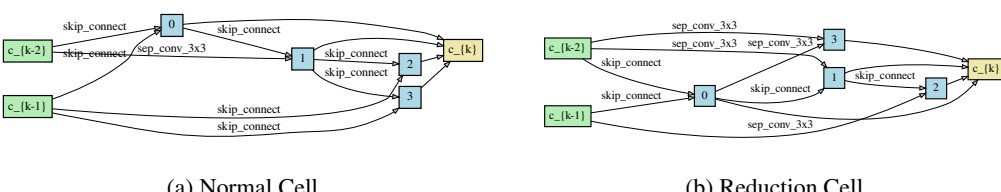

(a) Normal Cell                    (b) Reduction Cell

Figure 6: Normal and reduction cells found by DARTS on CIFAR-10 in search space S1.

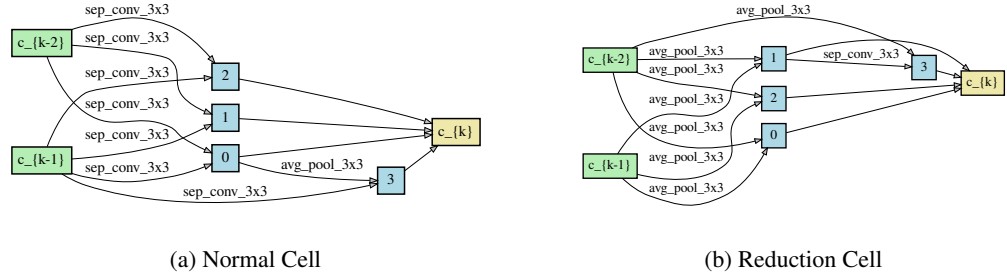

(a) Normal Cell

(b) Reduction Cell

Figure 7: Normal and reduction cells found by DARTS on CIFAR-10 in search space S2.

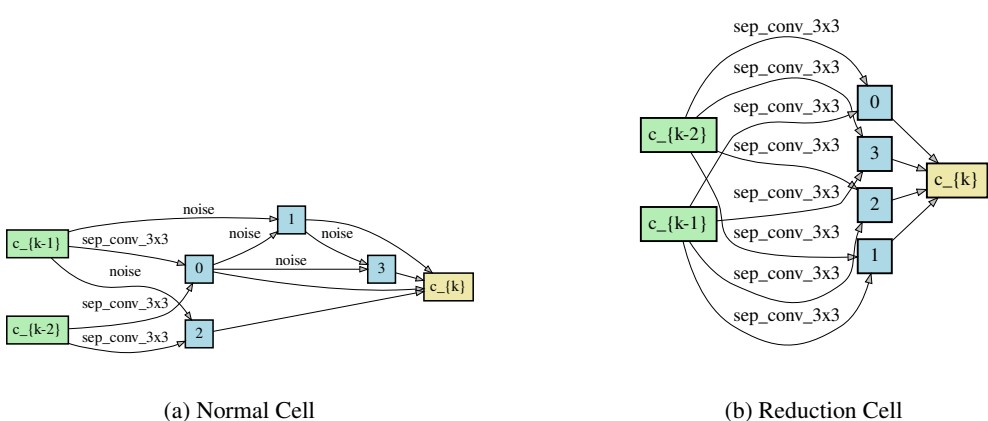

(a) Normal Cell

(b) Reduction Cell

Figure 8: Normal and reduction cells found by DARTS on CIFAR-10 in search space S3.

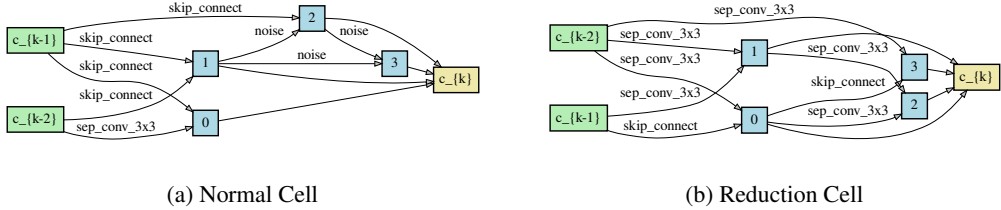

(a) Normal Cell

(b) Reduction Cell

Figure 9: Normal and reduction cells found by DARTS on CIFAR-10 in search space S4.

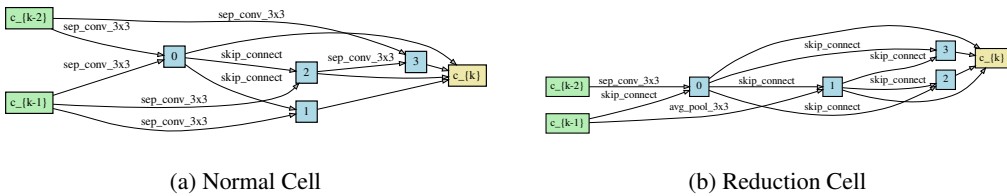

(a) Normal Cell

(b) Reduction Cell

Figure 10: Normal and reduction cells found by DARTS on CIFAR-10 in search space S5.

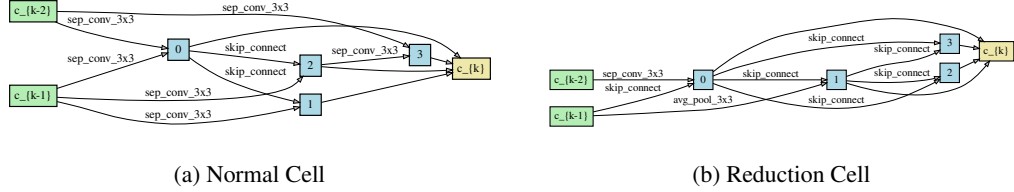

(a) Normal Cell

(b) Reduction Cell

Figure 11: Normal and reduction cells found by DARTS on CIFAR-10 in search space S6.

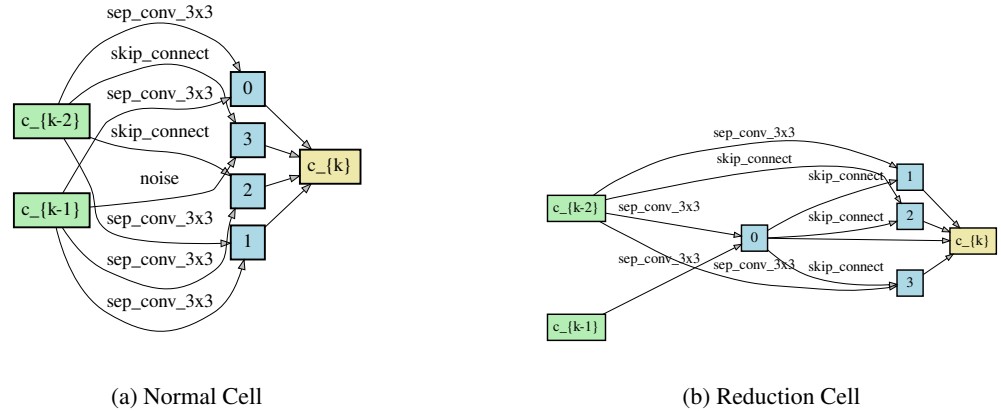

(a) Normal Cell

(b) Reduction Cell

Figure 12: Normal and reduction cells found by DARTS on CIFAR-10 in search space S7.

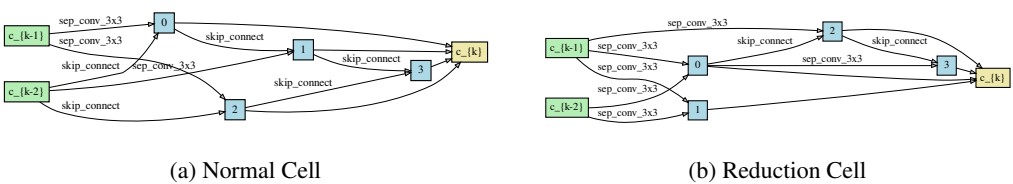

(a) Normal Cell

(b) Reduction Cell

Figure 13: Normal and reduction cells found by DARTS on CIFAR-100 in search space S1.

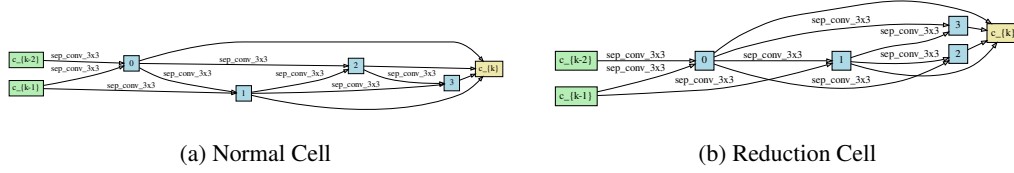

(a) Normal Cell

(b) Reduction Cell

Figure 14: Normal and reduction cells found by DARTS on CIFAR-100 in search space S2.

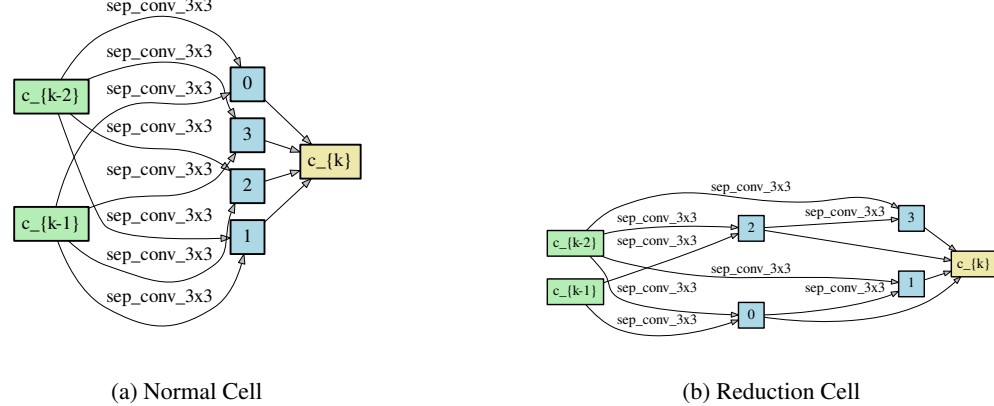

(a) Normal Cell                    (b) Reduction Cell

Figure 15: Normal and reduction cells found by DARTS on CIFAR-100 in search space S3.

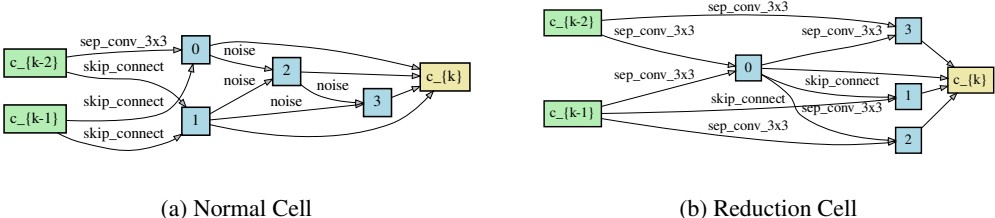

(a) Normal Cell                    (b) Reduction Cell

Figure 16: Normal and reduction cells found by DARTS on CIFAR-100 in search space S4.

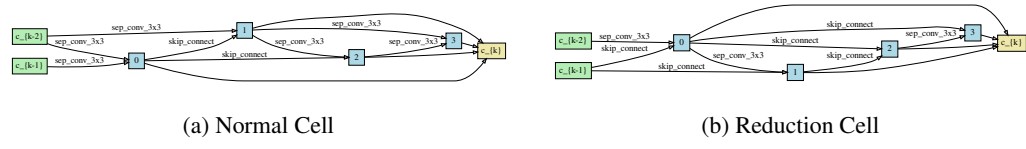

(a) Normal Cell                    (b) Reduction Cell

Figure 17: Normal and reduction cells found by DARTS on CIFAR-100 in search space S5.

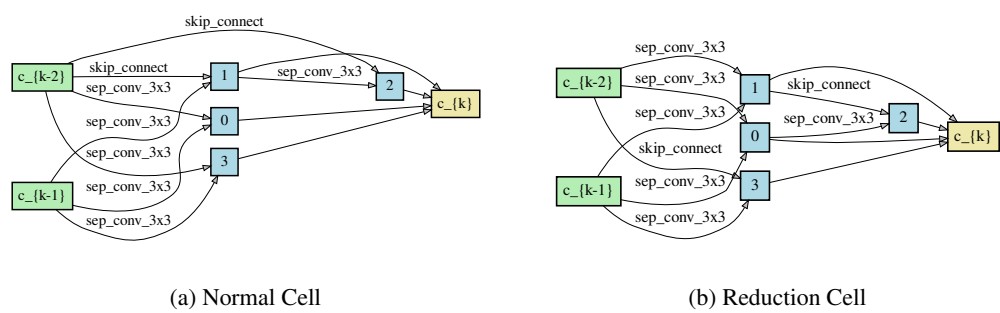

(a) Normal Cell                    (b) Reduction Cell

Figure 18: Normal and reduction cells found by DARTS on CIFAR-100 in search space S6.

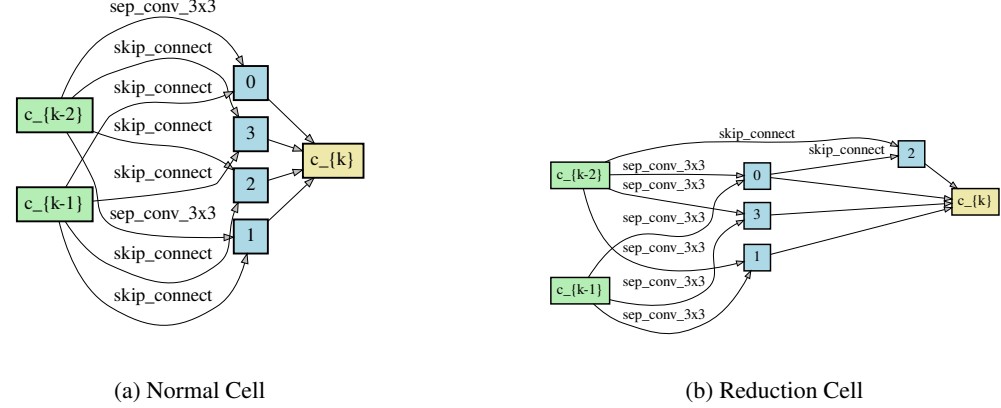

(a) Normal Cell                     (b) Reduction Cell

Figure 19: Normal and reduction cells found by DARTS on CIFAR-100 in search space S7.

# B   Gradient Distance Similarity Measures

In Section 5.1, four gradient distance measures are present. Next, we provide a comprehensive mathematical definition for each measure. Let $g^o_{train}$ and $g^o_{val}$ denote the gradients of the operation parameters $P_o$ on a training batch and a validation batch, respectively.

The GM score with $L_2$ distance can be formulated as follows:

$$GM_{L_2}(g^o_{train}, g^o_{val}) = \frac{\sum_{m=1}^{M} \|g^o_{train} - g^o_{val}\|_2}{M}. \tag{2}$$

And the GM score with direction distance can be formulated as follows:

$$GM_{direction}(g^o_{train}, g^o_{val}) = \frac{\sum_{m=1}^{M} \frac{\|g^o_{train} \odot g^o_{val} < 0\|_1}{\|P_o\|_0}}{M}, \tag{3}$$

where $\odot$ denotes the element-wise product, and $\|g^o_{train} \odot g^o_{val} < 0\|_1$ represents the number of elements whose directions are different.

However, the hard threshold method does not use the averaged gradient distances over $M$ steps to calculate the GM score. Instead, it directly checks whether the direction distance can consistently exceed a threshold over the consecutive $M$ steps. Let $\xi$ denote the threshold. The hard threshold method checks whether $\frac{\|g^o_{train} \odot g^o_{val} < 0\|_1}{\|P_o\|_0} > \xi$ can still be satisfied for $M$ consecutive steps.

# C   Supplementary Experimental Results

## C.1   Quantitative Analysis on Negative Correlation between Validation Loss and Architecture Parameter

The results of our motivation experiments, as depicted in Figure 1, indicate a negative correlation between validation loss and architecture parameter. To further provide a more rigid analysis, we calculate the Kendall-correlations between validation loss and architecture parameter for each operation and present the results in the Table 9. It can be observed that there is a clear negative correlation between validation losses and architecture parameters, particularly for parametric operations.

Table 9: Kendall-$\tau$ correlations for architecutre parameters and validation losses of different operations in Figure 1a.

| Skip-Connections | Sep-Conv-3x3 | Sep-Conv-5x5 | Dil-Conv-3x |
|:---:|:---:|:---:|:---:|
| $-0.36$ | $-0.59$ | $-0.73$ | $-0.72$ |

## C.2 Results with Longer Training Epoches

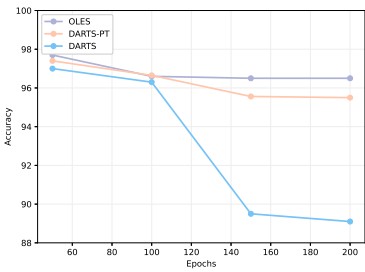

Figure 20: Test accuracy of searched architectures on CIFAR-10 with longer search epochs.

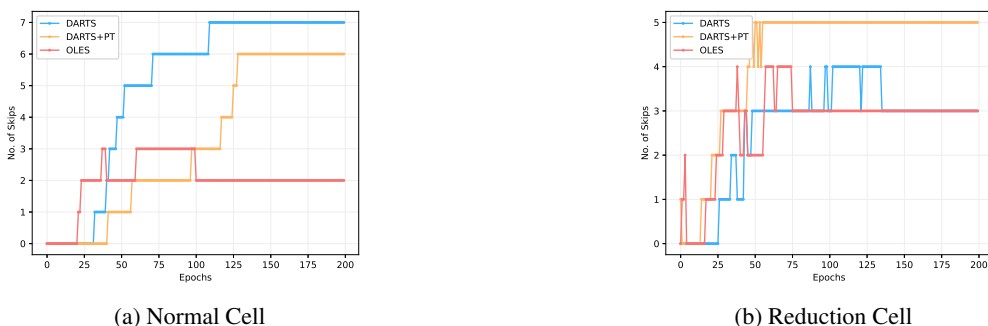

(a) Normal Cell          (b) Reduction Cell

Figure 21: The number of skip connections in searched architectures on CIFAR-10. In Figure 21b, there are some overlaps among the curves of DARTS and OLES.

Despite previous attempts to investigate the origins of the skip connection domination issue and its corresponding solutions, it remains an open problem. These methods have put forth their hypotheses but have not fully resolved the issue. To further exemplify the efficacy of addressing the skip connection domination issue, we increase the number of iterations of supernet training. As depicted in Figures 20 and 21, DARTS-PT Wang et al. [2021] fails to address the skip connection domination issue when extending to very long training epoches, whereas OLES remains stable.

## C.3 Hyperparameter Analysis

In OLES, the gradient matching score is dynamically computed by averaging over every 20 iterations throughout the entire training process. The threshold $\xi$ is determined based on the GM scores within the initial 20 iterations. We conduct experiments to assess various iterations along with their corresponding thresholds across different search spaces to explain the rationale behind selecting 20 iterations. The results, as shown in Table 10, demonstrate that computing the GM score using an average across every 20 iterations yields the highest accuracy.

## C.4 Results on DARTS S1-S4

DARTS S1-S4 are simplified variations of the DARTS search space proposed by R-DARTS Zela et al. [2019] to investigate DARTS' failure mode. In Appendix A, we have further expanded these search spaces and presented DARTS' failure mode (both in terms of performance and the searched architectures) in these extended search spaces. In Table 11, we additionally provide a performance comparison between OLES and other methods on DARTS S1-S4. The results demonstrate that OLES also consistently outperforms DARTS, as well as R-DARTS Zela et al. [2019] and DARTS-PT Wang et al. [2021] which are also dedicated to addressing the skip connection domination issue.

Table 10: Performance on various search spaces using different numbers of iterations to calculate the GM scores.

| Iterations | DARTS Search Space | | NAS-BENCH-201 Search Space | | |
| | CIFAR-10 | CIFAR-100 | CIFAR-10 | CIFAR-100 | ImageNet16-120 |
|---|---|---|---|---|---|
| 10 | 97.33 | 80.83 | 80.57 | 47.93 | 26.29 |
| 20 | **97.70** | **82.70** | **93.89** | **70.75** | **44.38** |
| 30 | 97.05 | 82.39 | 91.93 | 66.97 | 38.87 |
| 40 | 97.19 | 81.32 | 89.76 | 64.69 | 32.73 |

Table 11: Performance comparison on the DARTS S1-S4 search spaces.

| Datasets | Spaces | DARTS | R-DARTS | DARTS-PT | OLES |
|---|---|---|---|---|---|
| CIFAR-10 | S1 | 3.28 | 3.31 | 2.79 | **2.76** |
| | S2 | 2.55 | 2.44 | 2.47 | **2.43** |
| | S3 | 3.69 | 3.56 | 2.64 | **2.57** |
| | S4 | 3.05 | 3.05 | 2.92 | **2.73** |
| CIFAR-100 | S1 | 26.05 | 22.24 | 23.16 | **22.10** |
| | S2 | 25.40 | 23.34 | **22.10** | 22.18 |
| | S3 | 24.70 | 21.94 | 20.80 | **20.70** |
| | S4 | 21.35 | 20.70 | 19.98 | **19.18** |

## C.5 Comparison with Train-Free NAS Methods

Table 12 compares OLES with two recently proposed train-free NAS methods, TE-NAS Zhao et al. [2021] and ZiCo Li et al. [2023]. It is evident that OLES still outperforms train-free methods in most cases. While train-free methods exhibit slightly superior performance on NAS-Bench-201 due to its compact search space, in larger search spaces such as the DARTS search space, OLES performs better than TE-NAS and ZiCo. Train-free NAS is a highly promising research direction, as it not only greatly enhances NAS efficiency but also promotes a deeper understanding of neural networks.

Table 12: Comparison with TE-NAS and ZiCo. The test errors are reported.

| | NAS-Bench-201 | | | DARTS Search Space | | |
| | CIFAR-10 | CIFAR-100 | ImageNet-16-120 | CIFAR-10 | ImageNet | |
| | | | | | top-1 | top-5 |
|---|---|---|---|---|---|---|
| TE-NAS | 6.1±0.47 | 28.76±0.56 | 57.62±0.46 | 2.63±0.064 | 26.2 | 8.3 |
| ZiCo | 6.0±0.4 | 28.9±0.3 | 58.20±0.3 | 2.45±0.11 | - | - |
| OLES | 6.3±0.15 | 29.60±0.22 | **56.03 ± 0.38** | **2.41 ± 0.11** | 24.5 | 7.4 |

# D  Searched Architectures

## D.1  Architecture on MobileNet-like Search Space

Following the experimental settings in Chu et al. [2019a], we modify the MobileNetV2 Sandler et al. [2018] search space and perform OLES on ImageNet in this search space. The searched architecture is illustrated in Figure 22.

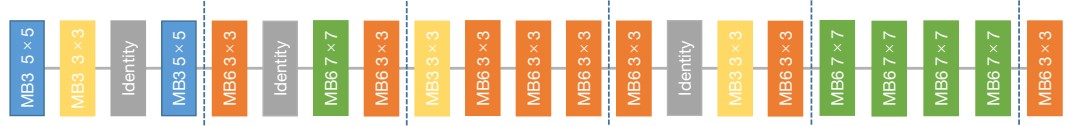

Figure 22: The architecture searched on the MobileNet-like search space.

## D.2 Training With Longer Epochs

We train OLES and several other differentiable architecture search methods with 200 epochs to verify whether they are robust enough in this case. According to our conjecture, as the number of training epochs increases, the operations in the supernet become more prone to overfitting, leading to a more serious issue of the domination of skip connections. In this section, we present some additional results of the searched architectures using different numbers of training epochs.

### D.2.1 Results on OLES

As described in Section 5.3, OLES can survive with longer training epochs. Specifically, the number of skip connections in OLES keeps stable even when the number of training epochs reaches 150 or 200. The corresponding architecture genotypes on CIFAR-10 and CIFAR-100 are visualized in Figures 23-26.

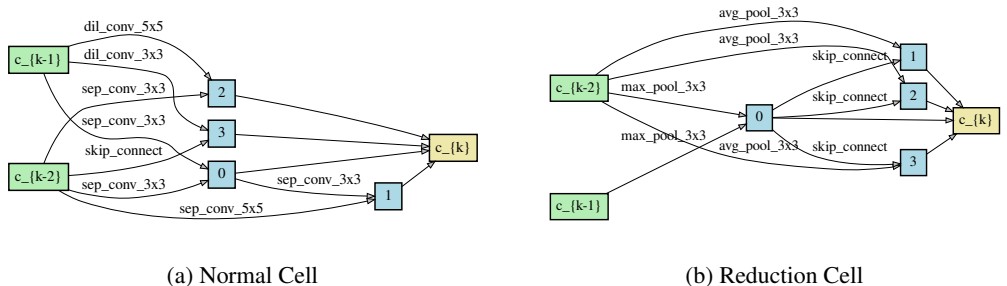

(a) Normal Cell                                       (b) Reduction Cell

Figure 23: Normal and Reduction cells discovered by OLES on CIFAR-10 for 150 epochs in the standard DARTS' search space.

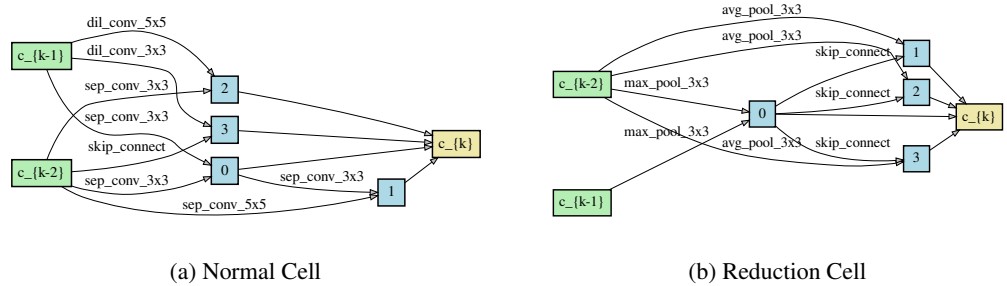

(a) Normal Cell                                       (b) Reduction Cell

Figure 24: Normal and Reduction cells discovered by OLES on CIFAR-10 for 200 epochs in the standard DARTS' search space.

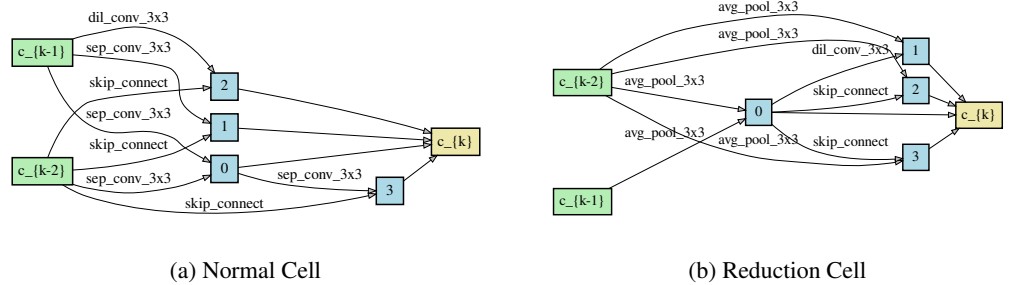

(a) Normal Cell                                    (b) Reduction Cell

Figure 25: Normal and Reduction cells discovered by OLES on CIFAR-100 for 150 epochs in the standard DARTS' search space.

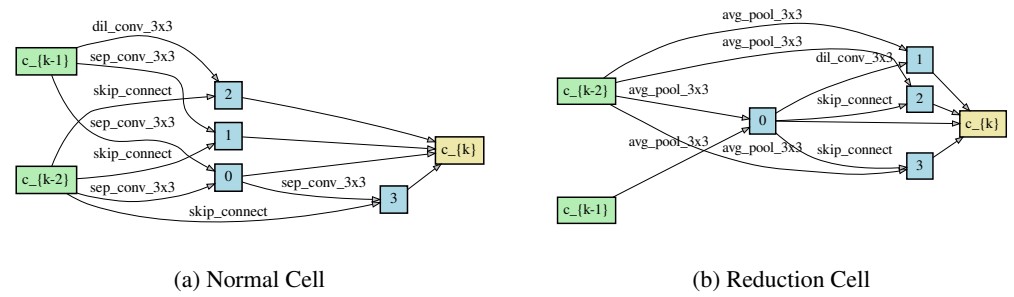

(a) Normal Cell                                    (b) Reduction Cell

Figure 26: Normal and Reduction cells discovered by OLES on CIFAR-100 for 200 epochs in the standard DARTS' search space.

### D.2.2 Results on FairDARTS

FairDARTS Chu et al. [2019a] adopts the *Sigmoid* instead of the exclusive *Softmax* for architecture parameters and selects operations through a threshold value. However, our experiments show that no operation can achieve the pre-defined threshold value in the normal cell when FairDARTS is trained for 150 or 200 epochs. Following FairDARTS' settings, we train it for 50, 150, and 200 epochs. Table 13 shows the architecture genotypes, which can be visualized in Figure 27 and Figure 28.

Table 13: Architecture genotypes found by FairDARTS on CIFAR-10 for 50, 150 and 200 epochs, respectively.

| Epochs | Architecture genotype |
|---|---|
| 50 | Genotype(normal=[('sep_conv_3x3', 2, 0), ('dil_conv_5x5', 2, 1), ('dil_conv_-3x3', 3, 0), ('dil_conv_3x3', 3, 1), ('sep_conv_5x5', 4, 1)], normal_concat = range(2, 6), reduce= [ ('skip_connect', 2, 0), ('dil_conv_5x5', 2, 1), ('skip_-connect', 3, 2), ('sep_conv_3x3', 3, 0), ('skip_connect', 4, 2), ('sep_conv_3x3', 4, 0), ('skip_connect', 5, 2), ('skip_connect', 5, 0)], reduce_concat=range(2, 6)) |
| 150 | Genotype(normal=[ ], normal_concat=range(2, 6), reduce=[('max_pool_3x3', 2, 0), ('max_pool_3x3', 2, 1), ('skip_connect', 3, 2), ('max_pool_3x3', 3, 0), ('skip_connect', 4, 2), ('max_pool_3x3', 4, 0), ('skip_connect', 5, 3), ('skip_-connect', 5, 2)], reduce_concat= range(2, 6)) |
| 200 | Genotype(normal=[ ], normal_concat=range(2, 6), reduce=[('avg_pool_3x3', 2, 0), (' max__pool_3x3', 2, 1), ('skip_connect', 3, 2), ('max_pool_3x3', 3, 0), ('skip_connect', 4, 3), ('skip_connect', 4, 2), ('sep_conv_3x3', 5, 0), ('dil_conv_-3x3', 5, 1)], reduce_concat= range(2, 6)) |

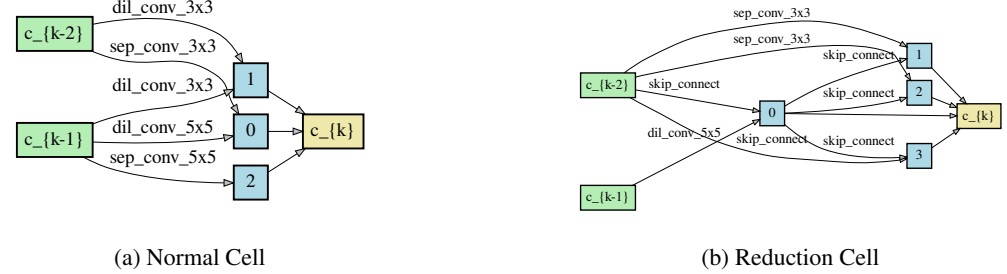

(a) Normal Cell            (b) Reduction Cell

Figure 27: Normal and Reduction cells discovered by FairDARTS on CIFAR-10 for 50 epochs in the standard DARTS' search space.

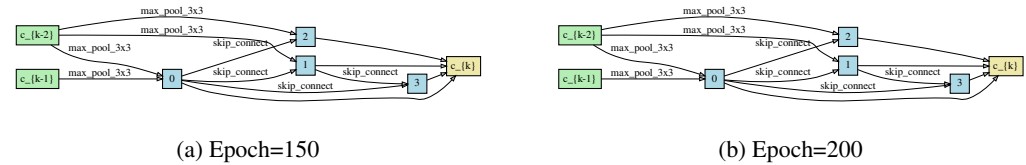

(a) Epoch=150            (b) Epoch=200

Figure 28: Reduction cells discovered by FairDARTS on CIFAR-10 for 150 and 200 epochs in the standard DARTS' search space. Note that normal cells are empty when FairDARTS is trained for 150 and 200 epochs.

### D.2.3 Results on DARTS+PT

DARTS+PT Wang et al. [2021] evaluates operation strength by measuring its contribution to the supernet's performance instead of the magnitude of the architecture parameter. Following DARTS+PT's settings, we train DARTS with the permutation-based architecture selection method for 50, 150, and 200 epochs, and the corresponding architecture genotypes are shown in Table 14. When extending to over 150 epochs, the resulting architectures are also full of skip connections. Furthermore, to avoid inaccurate evaluation of the operation strength during discretization, DARTS+PT needs to fine-tune the remaining supernet until it converges again. The architecture genotypes with tuned supernets are shown in Table 15, Figure 29, and Figure 30. It's worth noting that the test error of the final resulting architecture for 200 epochs on CIFAR-10 will drop to $6.28\%$.

Table 14: Architecture genotypes found by DARTS+PT on CIFAR-10 for 50, 150, and 200 epochs, respectively. (without supernet tuning)

| Epochs | Architecture genotype |
|---|---|
| 50 | Genotype(normal=[('sep_conv_3x3', 0), ('dil_conv_3x3', 1), ('sep_conv_3x3', 0), ('dil_conv_3x3', 2), ('dil_conv_3x3', 3), ('skip_connect', 0), ('dil_conv_5x5', 4), ('dil_conv_5x5',3)], normal_concat=range(2, 6), reduce=[('max_pool_3x3', 0), ('max_pool_3x3', 1), ('max_pool_3x3', 0), ('skip_connect', 2), ('skip_connect', 2), ('skip_connect', 3), ('skip_connect', 2), ('avg_pool_3x3', 0)], reduce_concat=range(2, 6)) |
| 150 | Genotype(normal=[('dil_conv_3x3', 1), ('sep_conv_3x3', 0), ('skip_connect', 2), ('skip_connect', 0), ('skip_connect', 1), ('skip_connect', 2), ('skip_connect', 1), ('skip_connect', 2)], normal_concat=range(2, 6), reduce=[('max_pool_3x3', 1), ('max_pool_3x3', 0), ('skip_connect', 2), ('max_pool_3x3', 0), ('skip_connect', 2), ('skip_connect', 3), ('skip_connect', 3), ('skip_connect', 2)], reduce_concat=range(2, 6)) |
| 200 | Genotype(normal=[('dil_conv_3x3', 1), ('sep_conv_3x3', 0), ('skip_connect', 2), ('skip_connect', 1), ('skip_connect', 1), ('skip_connect', 2), ('skip_connect', 1), ('skip_connect', 0)], normal_concat=range(2, 6), reduce=[('max_pool_3x3', 1), ('avg_pool_3x3', 0), ('skip_connect', 2), ('avg_pool_3x3', 0), ('skip_connect', 2), ('skip_connect', 3), ('skip_connect', 3), ('skip_connect', 2)], reduce_concat=range(2, 6)) |

Table 15: Architecture genotypes found by DARTS+PT on CIFAR-10 for 150 and 200 epochs, respectively. (with supernet tuning)

| Epochs | Architecture genotype |
|--------|----------------------|
| 150 | Genotype(normal=[('sep_conv_3x3', 0), ('dil_conv_3x3', 1), ('skip_connect', 0), ('skip_connect', 2), ('skip_connect', 0), ('dil_conv_5x5', 3), ('skip_connect', 1), ('sep_conv_5x5', 4)], normal_concat=range(2, 6), reduce=[('max_pool_3x3', 0), ('max_pool_3x3', 1), ('avg_pool_3x3', 0), ('skip_connect', 2), ('dil_conv_-5x5', 0), ('skip_connect', 3), ('dil_conv_5x5', 2), ('skip_connect', 4)], reduce_-concat=range(2, 6)) |
| 200 | Genotype(normal=[('sep_conv_3x3', 0), ('dil_conv_3x3', 1), ('sep_conv_3x3', 0), ('skip_connect', 2), ('skip_connect', 0), ('dil_conv_5x5', 3), ('skip_connect', 1), ('dil_conv_5x5', 4)], normal_concat=range(2, 6), reduce=[('sep_conv_3x3', 0), ('max_pool_3x3', 1), ('max_pool_3x3', 0), ('skip_connect', 2), ('sep_conv_-3x3', 1), ('skip_connect', 3), ('sep_conv_5x5', 0), ('skip_connect', 3)], reduce_-concat=range(2, 6)) |

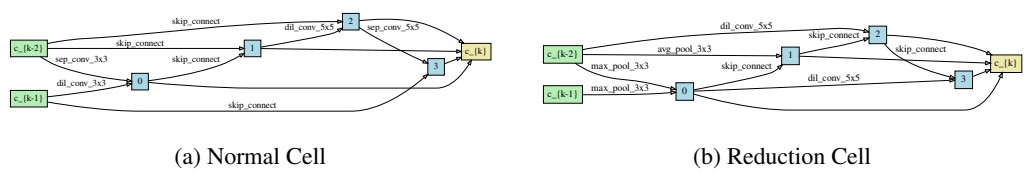

(a) Normal Cell        (b) Reduction Cell

Figure 29: Normal and Reduction cells discovered by DARTS+PT on CIFAR-10 for 150 epochs in the standard DARTS' search space (with supernet tuning).

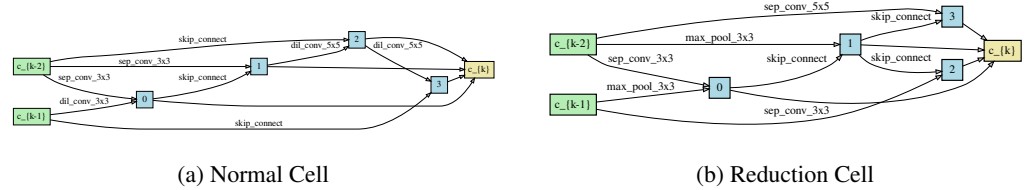

(a) Normal Cell        (b) Reduction Cell

Figure 30: Normal and Reduction cells discovered by DARTS+PT on CIFAR-10 for 200 epochs in the standard DARTS' search space (with supernet tuning).

