# OpenReview forum: "Operation-Level Early Stopping for Robustifying Differentiable NAS"
_NeurIPS.cc/2023/Conference — NeurIPS 2023 poster_

### Official Review · Reviewer_nZNk · 2023-07-06

**Soundness:** 3 good
**Presentation:** 2 fair
**Contribution:** 2 fair
**Rating:** 5
**Confidence:** 3

**Summary:**

This paper studies the robustness issue of DARTS from the perspective of overfitting. It uses gradient matching scores to measure the overfitting issues, and proposes an early-stop strategy to address the problem of saturated skip connections in normal DARTS. The proposed approach has been evaluated on a number of search spaces, showing comparable results to the state of the art.

**Strengths:**

+ The idea of using similarity between gradient directions of training vs validation batch makes sense.
+ Extensive experiments on various search spaces and benchmarks.
+ Decent results on DARTS C10 space comparing to SOTA.

**Weaknesses:**

- Although the idea to use early stopping to robustify DARTS is interesting, the metric used in this paper is very similar to that in GM-NAS [1]. Therefore it seems to me that the novelty of this paper is somewhat discounted.
- It is good to see experiments on a number of search spaces and benchmarks. However, the results on DARTS S1-S4 space is missing.
- Some discussion on overfitting is not very clear. e.g. from Fig.1, it is difficult to see quantitively how the negative correlation between arch params and val loss: indeed the curvature of the lines are different, but it would be better to have some more rigid analysis.


[1] Generalizing Few-Shot NAS with Gradient Matching. Shoukang Hu*, Ruochen Wang*, Lanqing Hong, Zhenguo Li, Cho-Jui Hsieh, and Jiashi Feng. ICLR 2022.

**Questions:**

* How the proposed approach performs on DARTS S1-S4?
* It seems this approach doesn't work so well on MobileNet space, any possible reasons for that?
* How this approach may compare to the train-free methods, like TE-NAS and ZiCo?

---

> ### Author Rebuttal · Authors · 2023-08-10
>
> Thank you for the helpful and insightful review, which is very helpful for us to further improve this paper. Next, we will answer your questions one by one, and we hope this will improve your acceptance of the paper.
>
> **W1.**
> Thank you for pointing this out. We apologize for not providing a detailed explanation for GM-NAS in the related work due to space constraints. In reality, GM-NAS and our approach are entirely distinct lines of work. Please refer to the global rebuttal for further detail.
>
> We hope our answers address your concerns, and we will consider adding the differences to our revised version.
>
> **W2 \& Q1.**
> Many thanks for the valuable advice. DARTS S1-S4 are not standard search spaces. They are simplified variations of the DARTS search space proposed by R-DARTS to investigate DARTS' failure mode. In our supplementary material, we further expanded these search spaces and presented DARTS' failure mode (both in terms of performance and the searched architectures) in these extended search spaces. To address your concerns, in the following table, we provide an additional performance comparison between OLES and other methods on DARTS S1-S4.
> It is evident that OLES also consistently outperforms DARTS, as well as R-DARTS and DARTS-PT which are also dedicated to addressing the skip connection domination issue.
>
> | Datasets | Spaces | DARTS | R-DARTS | DARTS-PT | OLES |
> |---|---|---|---|---|---|
> | CIFAR-10 | S1 | 3.28 | 3.31 | 2.79 | **2.76** |
> |  | S2 | 2.55 | 2.44 | 2.47 | **2.43** |
> |  | S3 | 3.69 | 3.56 | 2.64 | **2.57** |
> |  | S4 | 3.05 | 3.05 | 2.92 | **2.73** |
> | CIFAR-100 | S1 | 26.05 | 22.24 | 23.16 | **22.10** |
> |  | S2 | 25.40 | 23.34 | 22.10 | 22.18 |
> |  | S3 | 24.70 | 21.94 | 20.80 | **20.70** |
> |  | S4 | 21.35 | 20.70 | 19.98 | **19.18** |
>
> **W3.**
> Thanks for the suggestion. In Figure 1, through the curves of validation losses and architecture parameters over search iterations, it is evident that there exists a negative correlation between validation loss and architecture parameter. To provide a more rigid analysis, we calculated the Kendall-$\tau$ correlations between validation loss and architecture parameter for each operation. The results are presented in the following table.
>
> | Skip-Connections | Sep-Conv-3x3 | Sep-Conv-5x5 | Dil-Conv-3x |
> |---|---|---|---|
> | -0.36 | -0.59 | -0.73 | -0.72 |
>
> It can be observed that there is a clear negative correlation between validation losses and architecture parameters, particularly for parametric operations, where the negative correlation is more noticeable.
>
> Furthermore, we simplify the scenario by considering only one set of candidate operations $\{o_1, o_2, o_3\}$.  Suppose their corresponding validation losses and architecture parameters are $\{l_1, l_2, l_3\}$ and $\{\alpha_1, \alpha_2, \alpha_3\}$, respectively. The overall validation loss can be expressed as:
> $$
> l = l_1 \cdot \frac{e^{\alpha_1}}{e^{\alpha_1} + e^{\alpha_2} + e^{\alpha_3}} + l_2 \cdot \frac{e^{\alpha_2}}{e^{\alpha_1} + e^{\alpha_2} + e^{\alpha_3}} + l_3 \cdot \frac{e^{\alpha_3}}{e^{\alpha_1} + e^{\alpha_2} + e^{\alpha_3}}
> $$
> then
> $$
> \frac{\partial l}{\partial \alpha_1} = \frac{(l_1 - l_2)e^{\alpha_1 \alpha_2} + (l_1 - l_3) e^{\alpha_1 \alpha_3}}{(e^{\alpha_1} + e^{\alpha_2} + e^{\alpha_3})^2}
> $$
> Hence, when $l_1$ is larger, the corresponding $\frac{\partial l}{\partial \alpha_1}$ will be larger as well, leading to a more significant decrease in $\alpha_1$ during gradient descent. Other operations are similar. Although this scenario is significantly straightforward, it illustrates some connections between validation losses and architecture parameters. We hope our answers address your concerns.
>
> **Q2.**
> Thanks for the comment. Indeed, in the MobileNet search space, our approach has achieved competitive performance compared to state-of-the-art methods, slightly trailing behind several more complex approaches. As the MobileNet search space is not naturally designed for DARTS, we had to make several modifications (ProxylessNAS has adapted to this search space, and FairDARTS did not open-source their search process on MobileNet). Therefore, we believe there is still room for optimization in the results presented. Here, the purpose of using the MobileNet search space is to demonstrate the effectiveness of our approach in different search spaces.
>
> Furthermore, our approach is remarkably simple, involving only minimal modifications to the original DARTS, thus incurring negligible additional overhead. It is easy to enhance the performance of OLES by integrating it with other more advanced differentiable NAS algorithms, conducting more fine-tuning, and leveraging advanced training tricks.
>
> **Q3.**
>
> Many thanks for the suggestion. Train-free NAS utilizes proxy metrics to predict the test performance or ranking of architectures without the need for training. However, current train-free methods still slightly lag behind traditional NAS methods in terms of performance, especially in large search spaces.
>
> In the following table, we compare OLES with TE-NAS and ZiCo. It is evident that OLES still outperforms train-free methods in most cases. Benefiting from the compact search space, train-free methods exhibit slightly superior performance on NAS-Bench-201. However, in large search spaces such as DARTS search space, OLES performs better than TE-NAS and ZiCo. Train-free NAS is a highly promising research direction, as it not only greatly enhances NAS efficiency but also promotes a deeper understanding of neural networks.
>
> || NAS-Bench-201 ||| DARTS Search Space |||
> |---|---|---|---|---|---|---|
> |||||| ImageNet |||
> || CIFAR-10 | CIFAR-100 | ImageNet-16-120 | CIFAR-10 | top-1 | top-5 |
> | TE-NAS | 6.1 $\pm$ 0.47 | 28.76 $\pm$ 0.56 | 57.62 $\pm$ 0.46 | 2.63 $\pm$ 0.064 | 26.2 | 8.3 |
> | ZiCo | 6.0 $\pm$ 0.4 | 28.9 $\pm$ 0.3 | 58.20 $\pm$ 0.3 | 2.45 $\pm$ 0.11 | - | - |
> | OLES | 6.3 $\pm$ 0.15 | 29.60 $\pm$ 0.22 | **56.03 $\pm$ 0.38** | **2.41 $\pm$ 0.11** | **24.5** | **7.4** |

---

### Official Review · Reviewer_5e2k · 2023-07-06

**Soundness:** 2 fair
**Presentation:** 4 excellent
**Contribution:** 1 poor
**Rating:** 4
**Confidence:** 5

**Summary:**

The authors are adressing the issue of converging to a degenerated solution (many skip connections) using DARTS. The authors connect this behavior to an overfitting to the train data. To remedy this issue, they suggest to apply early stopping based on the correlation of gradients during the architecture parameters training and the wekths parameters training.

**Strengths:**

The paper is trying to tackle an interesting problem in an elegant way. The story is clearly stated and the introduced approach is a nice extension of the classical early stopping. The performance of this method is on par or better than many other more sophisticated (and to some extent complicated) method.

**Weaknesses:**

* The authors are claiming that the dominance of skip connections is due to the quick overfitting of weighted operations to the train data. The whole story of the paper relies on this assumption which is unfortunately not sufficiently supported. While curves on figure 1(a) are supporting this claim, figures 3(a) and 3(b) are a bit confusing.  I would have expected the test performance to drop for both cases.

* There is an issue regarding the reported numbers: the authors are claiming that their method benefits operations with parameters and the test error is dropping (as expected), I am wondering why the number of parameters is barely different.

**Questions:**

Adressing the mentioned weaknesses appropriatly would lead to a higher rating

---

> ### Author Rebuttal · Authors · 2023-08-10
>
> Thanks a lot for your considerate feedback. We sincerely appreciate your engagement in the review. Next, we will address your concerns one by one, and we hope this will improve your view of the paper.
>
> **W1.**
> Thanks a lot for your careful and insightful observation. In fact, in Figures 3(a) and 3(b), the number of iterations of DARTS has not yet reached the threshold where the test performance starts to drop, indicating that the overfitting issue has not yet appeared. In Figure 4 of our attached PDF, we have redrawn Figures 3(a) and 3(b) and extended the number of training iterations. Consequently, it becomes apparent that when the number of training iterations continuously increases (>50), DARTS exhibits a noticeable decline in test performance. Moreover, Figure 3(c) in the original paper further corroborates this finding, as the problem of performance collapse due to the skip connection domination issue more easily appears in NAS-Bench-201. As a result, it manifests the test performance declines at an earlier stage. We hope our answers address your concerns, and we will consider improving Figures 3(a) and 3(b) in the revised version.
>
> **W2.**
> Thanks for pointing this out. Our approach aims to address the issue of performance collapse caused by the abnormal aggregation of skip connections in differentiable NAS. NAS methods fundamentally seek the optimal model architecture. In reality, models with more parameters are not necessarily superior to those with fewer parameters. The phenomenon you described, i.e., that the number of parameters in OLES is barely different with other methods, actually serves to illustrate that OLES can resolve the issue of abnormal skip connection domination, rather than invariably benefit operations with parameters.
>
> Furthermore, in our comparisons, we presented the optimal results of other methods (mostly from the original paper's reported results). These results were obtained before the occurrence of skip connection domination.  Once the skip connection domination issue emerges, there will be a noticeable performance collapse. As shown in Figures 1&2 in our attached PDF and Section C.2 in our supplementary material and, DARTS and other methods like DARTS-PT encounter skip connection domination and performance collapse after training for longer epochs.  Thus, there are no significant differences in the number of parameters.
>
> We hope our answers address your concerns. Thanks again for this insightful comment.

---

> > ### Author Response · Authors · 2023-08-21
> >
> > We would like to know if our responses have adequately addressed your concerns or if further clarification is needed. If you find it appropriate, we appreciate your new rating. We are grateful for your time and thoughtful evaluation of our work.

---

### Official Review · Reviewer_nDad · 2023-07-07

**Soundness:** 3 good
**Presentation:** 3 good
**Contribution:** 3 good
**Rating:** 6
**Confidence:** 4

**Summary:**

This paper demonstrates the fundamental reason for the domination of skip connections in DARTS from the new perspective of overfitting of operations in the supernet, using preliminary experiments,  and proposed the operation-level early stopping method to mitigate this phenomenon by using the GM score metric during the searching.

**Strengths:**

The idea of using GM metric to decide whether to update the OPs is broadly used in efficient-training papers.  This paper proposes to use this metric to early stop the updating of specific OPs during the NAS procedure, based on their novel overfitting observations.

The comparison is intensive, showing the superiority of this proposed method, in terms of time cost, and accuracy.

**Weaknesses:**

1. The accuracy metric is one of the metrics to measure a NAS method. We also consider the Kendall rank correlation coefficient. Please compare with previous methods using this metric, because I am not sure whether the proposed early stopping mechanism will hurt the ranking or not. Usually, we say the ranking performance of a NAS method may be more important than the accuracy of the searched model.

2. About the overfitting threshold, "we determine the threshold by averaging the cosine similarity over 20 iterations for 30 randomly initiated architectures in each search space"    Is the initial 20 iterations ok or enough for determining the threshold? The initial stages may have dramatic changes in gradient. Also, the threshold is fixed during the NAS procedure, shouldn't it be adaptive or scheduled?

**Questions:**

See Weakness

**Limitations:**

Yes, the authors adequately addressed the limitation.

---

> ### Author Rebuttal · Authors · 2023-08-10
>
> Thank you for the helpful and insightful review. Next, we will answer your questions one by one, and we hope this will improve your acceptance of the paper.
>
> **W1.**
> Thanks for the valuable suggestion. As shown in Figure 3 in our attached PDF, we compare the Kendall rank correlation coefficients for OLES and DARTS. The results indicate that the early stopping mechanism does not hurt the ranking performance of DARTS. In the following table, we also conduct a comparison with other NAS methods, including RandomNAS, DARTS, GDAS, and SGAS. (RandomNAS and GDAS sample 4 excellent architectures in 4 rounds, and other methods randomly select 10 architectures from a single run.)
>
> | RandomNAS$^*$ | DARTS | GDAS$^*$ | SGAS | OLES |
> |---|---|---|---|---|
> | 0.0909 | 0.13 | -0.1818 | 0.42 | 0.41 |
>
> Notably, the Kendall coefficient of OLES closely aligns with SGAS. SGAS [1]  aims to alleviate the effect of the degenerate search-retraining correlation problem. The results demonstrate that by mitigating operation parameter overfitting, differentiable NAS could focus on the potential of architectures themselves, thus enhancing the correlation between search metrics and the architectures discovered. We hope our answers address your concerns, and we will consider adding the comparison of the Kendall rank correlation coefficients into the revised version.
>
> **W2.**
> Thank you for pointing this out. Instead of relying on search or empirical knowledge to determine the early stopping threshold, we intend to employ an adaptive approach to determining this critical hyperparameter. It is important to clarify that our gradient matching (GM) score is dynamically computed by averaging over every 20 iterations throughout the entire training iterations, rather than being limited to the initial 20 iterations. We apologize for the confusion here. The selection of 30 randomly initiated architectures is carried out to initiate the experiment 30 times, enabling the identification of the optimal iteration number. As you pointed out, the initial stages may have dramatic changes in gradient values. Therefore, in our experiments, we compute the GM score throughout the entire training process to ensure the effectiveness of early stopping.
>
> To elucidate the rationale behind selecting 20 iterations, we conduct experiments to assess various iterations along with their corresponding thresholds across different search spaces. As shown in the following table, the accuracy attained through GM score computation over every 20 iterations achieves the highest accuracy.
>
> We have not yet found a better method to determine the threshold in a more adaptive or schedulable manner, and we will leave it for future work. We sincerely appreciate your valuable suggestions and we will make more clear introduction about the selection of  the overfitting threshold in our revised version.
>
> | Iterations |DARTS Search Space|| NAS-Bench-201 Search Space |||
> |---|---|---|---|---|---|
> || CIFAR-10 | CIFAR-100 | CIFAR-10 | CIFAR-100 | ImageNet-16-120 |
> | 10 | 97.33 | 80.83 | 80.57 | 47.93 | 26.29 |
> | 20 | **97.70** | **82.70** | **93.89** | **70.75** | **44.38** |
> | 30 | 97.05 | 82.39 | 91.93 | 66.97 | 38.87 |
> | 40 | 97.19 | 81.32 | 89.76 | 64.69 | 32.73 |
>
>
> [1]  Snas: stochastic neural architecture search, ICLR 2019.

---

> > ### Comment · Reviewer_nDad · 2023-08-10
> > **Thank you for the detailed feedback**
> >
> > Thank you for the detailed feedback, the rebuttal addressed my concerns. I would like to increase the score from borderline accept to weak accept.

---

> > > ### Author Response · Authors · 2023-08-10
> > > **Thank you for your thorough review and raising the score**
> > >
> > > Thank you for your thorough review and raising the score for our submission. Your valuable feedback guided us in refining our experimental approach and the suggestions guided us in providing more concise and coherent explanations.
> > > Thank you for your time, expertise, and for acknowledging our commitment to enhancing the quality of our research. We look forward to further enriching our work based on your insightful input.

---

### Official Review · Reviewer_XYdd · 2023-07-10

**Soundness:** 3 good
**Presentation:** 4 excellent
**Contribution:** 3 good
**Rating:** 7
**Confidence:** 4

**Summary:**

The paper focuses on the robustness issues in differentiable NAS, specifically the domination of skip connections. It first analyzes the issue from a novel perspective, proposing that the domination of skip connections arises due to the overfitting of operations in the supernet during training. Then, the paper proposes the operation-level early stopping method, which monitors each operation in the supernet and stops its training when it tends to overfit. The paper employs a gradient matching approach to detect overfitting, comparing the gradients' directions of operations on training and validation data. A significant deviation in direction is considered an indication of overfitting. The proposed OLES addresses the domination of skip connections with negligible additional overhead. Extensive experiments demonstrate the effectiveness of OLES on different datasets and search spaces.

**Strengths:**

S1. The paper demonstrates good originality. It provides a comprehensive analysis of the issue of the domination of skip connections in the differentiable NAS by dopting a new perspective. Although straightforward, the perspective makes sense and is interesting. Specifically, the paper aims to explain the cause of this issue through the overfitting of operations in the supernet. The paper proposes the operation-level early stopping (OLES) method, which introduces gradient matching to address this matter. OLES elegantly and effectively resolves the domination of skip connections, incurring negligible additional overhead.

S2. The experiments are thorough and well-organized, containing experiments in different search spaces and an in-depth analysis of the proposed algorithm. The empirical results demonstrate that the proposed OLES  achieves state-of-the-art performance on CIFAR. The availability of open-source codes further facilitates reproducibility.

S3. The presentation of ideas and algorithms is clear, while the references and background knowledge are comprehensive. The background knowledge and the issue to be solved are adequately introduced.

S4. The perspective and idea about the overfitting of operations have profound significance in uncovering the underlying causes of the domination of skip connections in DARTS. These novel perspectives, ideas, and algorithms contribute to a deep understanding of differentiable architecture search and may inspire future research in the field.

**Weaknesses:**

W1. It needs a thorough explanation of the used gradient matching method. The authors should provide a more detailed introduction to gradient matching and clarify the differences from other methods.

W2. While the proposed method demonstrates significant improvements over the original DARTS in the experiments, it does not have a competitive advantage compared to other state-of-the-art methods.

W3. The paper specifically focuses on addressing the domination of skip connections through operator-level early stopping. The authors are suggested to discuss how the concept of operator-level early stopping can be applied to other scenarios beyond differentiable architecture search.

**Questions:**

Q1. Are there any limitations to the methodology presented in this paper? Can it be applied to other differentiable NAS methods beyond the original DARTS?

Q2. Why does the proposed method not exhibit a competitive advantage over other state-of-the-art NAS methods?

---

> ### Author Rebuttal · Authors · 2023-08-10
>
> Thank you for the helpful and insightful review. We are glad to receive your positive response and acknowledgment of our work. Next, we will answer your questions one by one, and we hope this will improve your acceptance of our paper.
>
> **W1.**
> Thanks for the valuable advice. Gradient matching aims to leverage the gradient information of parameters to assist in making decisions and selections. For example, GM-NAS utilizes gradient matching scores to make splitting decisions, determining whether a module should be shared among child architectures. The gradient matching scores are computed based on the gradient information of different child models on shared parameters. There are also methods that utilize GM for dataset condensation.
> In our proposed OLES, we employ the GM score as an indicator for early stopping, preventing operation parameter overfitting.
> The gradient matching scores, in our approach, are calculated using the gradient information of parameters on both training and validation data. We hope our answers address your concerns.
>
> **W2 \& Q2.**
> Thank you for pointing this out. Compared to other sophisticated NAS methods, we aim to solve the skip connection domination issue in differentiable NAS and uncover the fundamental causes of this issue. OLES offers a novel perspective on the origins of the skip connection domination and proposes simple yet effective solutions. It is essential to note that our approach requires only minimal modifications to the original DARTS, thus incurring negligible additional overhead.
> Practitioners can easily enhance the performance of OLES by integrating it with other state-of-the-art differentiable NAS algorithms, conducting more fine-tuning, and leveraging advanced training tricks such as the SE module. We hope our answers address your concerns.
>
> **W3 \& Q1.**
> Many thanks for the suggestion. OLES can also be employed as a plugin module for other differentiable NAS methods. It is especially well-suited for scenarios where the NAS method alternately trains on the training and validation datasets. Depending on the specific differentiable NAS methods and search spaces, it may be necessary to adjust the early stopping threshold setting to achieve optimal performance.

---

> > ### Comment · Reviewer_XYdd · 2023-08-19
> > **Thank your for your response**
> >
> > Thank your for your response. I have no further questions.

---

> > > ### Author Response · Authors · 2023-08-19
> > > **Thanks very much for your feedback**
> > >
> > > Many thanks for your feedback. We are glad to hear that our response has addressed your concerns. Your evaluation and support on our work are greatly appreciated!  We are committed to integrating clarifications you suggested into the forthcoming permitted revision.

---

### Official Review · Reviewer_fTac · 2023-07-10

**Soundness:** 2 fair
**Presentation:** 3 good
**Contribution:** 2 fair
**Rating:** 3
**Confidence:** 4

**Summary:**


This paper propose a method namely operation-level early stopping to address the skip-connection domination issue in domain of differentiable architecture search (DARTS). Though this problem is heavily explored in the past, the authors believe that the key reason of skip-connection domination is because of they try to overfit the validation set used in DARTS based method, and their proposed operation-level early stop that conceptually stop the training of architecture parameter if the overfitting is observed.

**Strengths:**

The problem of skip-connection dominance is a long standing problem in DARTS domain, and novel method to address this has a clear motivation.
The hypothesis of overfitting is the root cause of skip-connection domination is novel and interesting.
Analysis of architecture with validation loss is quite clear and justify their hypothesis.
I appreciate the extensive and honest experiments on all kinds of settings, even though many of the results does not surpass the state-of-the-art.

**Weaknesses:**


I have several questions regarding this paper and hope to hear back from the authors.

1. Utilizing gradient matching as an indicator to perform early stop seems okay, but this paper lacks of sufficient analysis of what is the key difference between GM+DARTS and their approach, especially regarding why OLES is better. In the related work section, it only describes GM-NAS introduces gradient match score into NAS literature, but this is not enough to let readers understand the difference.
2. In essence, use early stop to avoid overfitting in DARTS is novel, but experiments seem to show that this OLES does not surpass previous DARTS+PT, which is another indicator to select DARTS operation. I do not understand the urgency to accept another indicator work with similar performance. In addition, similar as above, I do not see much comparison of how OLES surpass DARTS+PT. I agree that OLES can surpass original DARTS, but if it cannot surpass other methods that aims to address the skip-connection domination issue, this is inadequate to appear in a top-tier conference in my humble opinion.
3. In general, the experiments compared to state-of-the-art is inferior.
To the best of my knowledge, this approach has a close relationship with GM+DARTS and DARTS+PT, where in all experiments, the authors should compare to. However, for example, in Table 2, none of these methods exists. In addition, when solely compared to GM+DARTS, in Table 1, it is 0.05 test error better than GM+DARTS. In table 3, GM+DARTS is basically identical as OLES (24.5 v.s. 24.5). In Table 4, OLES surpass GM+DARTS by a margin of 0.2, which is kind of significant. However, in Table 5, GM + ProxylessNAS surpasses OLES again. Why we should accept a paper using the exact gradient matching score in a different manner, which seems inferior than the original approach?

In addition, I would like to see how this OLES address skip-connection domination with other approaches that aim the same target, not in terms of their final performance, but with respect to their performance to address skip-connection domination issue. After all, this is not the first paper aiming to address it.


**Questions:**

Same as above

**Limitations:**

Moderately discussed.

---

> ### Author Rebuttal · Authors · 2023-08-10
>
> Thank you for the insightful review, which is very helpful for us to further improve this paper. Next, we will answer your questions one by one, and we hope this will help to address your concerns and improve your view of the paper.
>
> **W1.**
> Thank you for pointing this out. We apologize for not providing a detailed explanation for GM-NAS in the related work due to space constraints. Although, as you mentioned, both methods employ the GM Score as a mathematical tool, GM-NAS and our approach are entirely distinct lines of works. Please refer to the global rebuttal for further detail.
>
> We hope our answers address your concerns. Thanks again for the comment, and we will consider adding the differences to our revised version.
>
> **W2.**
> Thanks for the comment. Actually, our approach almost consistently outperforms DARTS-PT and other NAS methods that address the skip connection domination issue. For your convenience, we highlight the comparison between our approach and other NAS methods in the following table. And in most cases, our method remains competitive compared to the SOTA results.
>
> | NAS method | CIFAR-10 | CIFAR-100 (Transfer) | CIFAR-100 | ImageNet|
> |---|---|---|---|---|
> | P-DARTS | 2.50 | 16.55 | 17.46 | 24.4 |
> | R-DARTS(L2) | 2.95 $\pm$ 0.21 | - | 18.24 | - |
> | FairDARTS | 2.54 | - | - | 24.9 |
> | DARTS- | 2.59 $\pm$ 0.08 | - | 17.51 $\pm$ 0.25 | 23.8 |
> | $\beta$-DARTS | 2.53 $\pm$ 0.08 | 16.24 $\pm$ 0.22 | 17.33 | 23.9 |
> | DARTS+PT | 2.48(2.61 $\pm$ 0.08) | 19.05 | 18.78 | 25.5 |
> | **OLES** | **2.30(2.41 $\pm$ 0.11)** | 16.30(16.35 $\pm$ 0.05) | **17.30** | 24.5 |
>
> Despite existing efforts to explore the origins of the skip connection domination issue and the corresponding solutions, it remains an open problem. These methods have put forth their hypotheses but have not fully resolved the issue. Compared to existing methods, the main contribution of our approach is that we offer a novel perspective on the origins of the skip connections domination and propose an effective solution. It is essential to note that our approach is remarkably simple, involving only minimal modifications to the original DARTS, thus incurring negligible additional overhead. We believe that the current performance of OLES is sufficient to validate our hypothesis and demonstrate that our method effectively resolves the skip connection domination issue in differentiable NAS.
>
> To further demonstrate the effectiveness of addressing the skip connection domination issue, we increase the number of iterations of supernet training. As depicted in Figures 1&2 in our attached PDF, DARTS-PT fails to solve the skip connection domination issue, indicating that their hypotheses do not entirely reveal the fundamental causes of this problem, or rather, they merely delay the occurrence of skip connection domination.
>
> Therefore, our work is meaningful as we offer a novel perspective and idea that not only proves effective but is also simpler and more straightforward to comprehend than previous perspectives. Instead of introducing yet another indicator, we aspire to stimulate further exploration in this field based on our insightful observations regarding the overfitting phenomenon. We believe this can lead to a deeper understanding of the differentiable architecture search process. We hope our answers address your concerns, and we will highlight the differences between our approach and other methods that focus on the skip-connection domination issue.
>
> **W3.**
> Thank you for pointing this out. For both GM-NAS and DARTS-PT, the results on CFIAR-100 are not found in their original papers. For more comprehensive comparisons, we reproduced the results of these two methods on CFIAR-100 using their open-source codes, and the results are also displayed in the table below.
>
> ||DARTS+PT|GM+DARTS|OLES|
> |---|---|---|---|
> | CIFAR-100 (Transfer) | 19.05 | 16.45 | **16.30** |
> | CIFAR-100 |18.78 | 17.42 | **17.30** |
>
> It can be observed that the proposed OLES performs better than GM+DARTS and DARTS+PT on CIFAR-100. As previously stated, GM-NAS is a much more complex method that falls in a distinct line of research. Due to its few-shot nature, it involves higher time and space complexities. Moreover, GM-NAS is orthogonal to our approach. OLES focuses on addressing the skip connections domination issue in differentiable NAS. We aim to uncover the fundamental causes of this issue and require only minimal modifications to the original DARTS, incurring negligible additional overhead. Despite this, our method remains highly competitive compared to GM-NAS. Furthermore, our main contribution is the insightful observation and hypothesis regarding overfitting, as well as the proposal of the early stopping method. The GM score, in this context, serves merely as an indicator for early stopping.
>
> To demonstrate the effectiveness of our approach in different search spaces, we also perform OLES on the MobileNet search space. As it is not naturally designed for DARTS, we had to make several modifications (ProxylessNAS has adapted to this search space, and FairDARTS did not open-source their search process on MobileNet). Therefore, we believe there is still room for optimization in the results presented. Nevertheless, OLES achieved highly competitive performance in the MobileNet search space.
>
> **W4.**
> Thanks for the suggestion. Besides the final performance, we display the number of skip connections in the searched architecture as the number of training iterations increases in Figure 2 of our attached PDF. It is evident that DARTS exhibits a severe skip connection domination issue, while OLES effectively resolves this problem. When training with an extended number of training epochs. Methods like DARTS-PT tend to have an increasing trend of skip connections, whereas OLES remains stable, which further verifies the insightful observations on the fundamental causes of the skip connection domination issue and the effectiveness of the proposed solution.

---

### Author Rebuttal · Authors · 2023-08-10

Dear AC and reviewers,

We would like to thank all the reviewers for their great efforts, insightful comments, and valuable suggestions, which are very helpful for us to further improve this paper. To the best of our efforts, we’ve diligently tried to address all the specific comments, including the minor ones that have been raised by each reviewer, by adding additional experiments and detailed analyses. Next, we would like to respond to the questions raised by the reviewers point by point, and we will incorporate these clarifications in the future version of our paper. We sincerely hope that our responses can address reviewers' concerns and translate into higher scores.

In the attached PDF, we present some additional experimental results to provide further clarification for addressing the concerns of the reviewers. In Figure 1, we compare the performance of our OLES against other methods when training for longer epochs. Similarly, Figure 2 displays the number of skip connections obtained by different methods. In Figure 3, we present the search-retraining Kendall coefficients of OLES and DARTS. Lastly, Figure 4 shows the results of the original paper's Figure 1(a) and Figure 1(b) under extended training iterations. We kindly recommend the esteemed reviewers refer to the content provided in the attached PDF while reviewing our rebuttal.

Here we provide general answers to similar questions mentioned by the reviewers.

**1. Distinctions between OLES and GM-NAS**

In reality, GM-NAS and our approach are entirely distinct lines of work, addressing completely different problems. They just both employ the GM Score as a mathematical tool. Specifically, GM-NAS falls within the field of Few-Shot NAS research. They argue that due to coupled optimization between child architectures caused by weight-sharing, One-Shot supernet’s performance estimation could be inaccurate, leading to degraded search results. As a result, they propose to reduce the level of weight-sharing by splitting the One-Shot supernet into multiple separated sub-supernets. GM-NAS utilizes gradient matching scores to make splitting decisions, determining whether a module should be shared among child architectures. The gradient matching scores are computed based on the gradient information of different child architectures on shared parameters.

In contrast, the proposed OLES aims to address the problem of skip connection donimation from a totally new perspective and employ the GM score as an indicator for early stopping, preventing operation parameter overfitting. The gradient matching scores, in our approach, are calculated using the gradient information of parameters on training and validation data.

Also, while GM-NAS offers performance enhancements to some extent, the time complexity of GM-NAS is larger, leading to higher search costs compared to OLES. As shown in Table 1 and Table 3, the search costs of GM-DARTS is 3 times and 6 times higher than OLES on CIFAR-10 and ImageNet, respectively. Due to only minimal modifications to the original DARTS, our approach incurs negligible additional overhead.

Furthermore, due to different lines of works, our approach and GM-NAS are orthogonal, enabling their combined usage to further enhance performance. We will try it in our future work.

Thanks again to all the reviewers for their constructive suggestions. We will do our best to improve our paper and address all concerns raised. We sincerely hope our answers can address the all the concerns reviewers have raised and improve the score of our paper.

Best regards,

Authors of Submission 11229

---

### Decision · Program_Chairs · 2023-09-21

**Decision:**

Accept (poster)

**Comment:**

This paper addresses the robustness issues in differentiable NAS via a new operation-level early stopping to address the skip-connection domination issue in domain of DART. Experiments show the effectiveness of OLES on different datasets and search spaces and it resolves the domination of skip connections with negligible additional overhead. Reviewer fTac questions the difference between GM+DARTS and OLES. The general rebuttal gives detailed and convincing comparison. Reviewers' concerns about experiments have been responded such as, OLES does not surpass previous DARTS+PT, which is wrong understanding that has been clarified in W2. Reviewer 5e2k raised the issues about reported numbers and test performance, which have been addressed by authors. Both Reviewer fTac and Reviewer 5e2k are not involved in the discussion. From Area chair's understanding, the authors have addressed the problems. Thus the whole paper is good enough for Neurips.